# Machine learning aided design of single-atom alloy catalysts for methane cracking

Jikai Sun[1,2], Rui Tu[1,2], Yuchun Xu[1], Hongyan Yang[1], Tie Yu ●[1] ✉, Dong Zhai ●[1], Xiuqin Ci[1] & Weiqiao Deng ●[1] ✉

The process of $CH_4$ cracking into $H_2$ and carbon has gained wide attention for hydrogen production. However, traditional catalysis methods suffer rapid deactivation due to severe carbon deposition. In this study, we discover that effective $CH_4$ cracking can be achieved at 450 °C over a Re/Ni single-atom alloy via ball milling. To explore single-atom alloy catalysis, we construct a library of 10,950 transition metal single-atom alloy surfaces and screen candidates based on C−H dissociation energy barriers predicted by a machine learning model. Experimental validation identifies Ir/Ni and Re/Ni as top performers. Notably, the non-noble metal Re/Ni achieves a hydrogen yield of 10.7 $gH_2$ $gcat^{-1}$ $h^{-1}$ with 99.9% selectivity and 7.75% $CH_4$ conversion at 450 °C, 1 atm. Here, we show the mechanical energy boosts $CH_4$ conversion clearly and sustained $CH_4$ cracking over 240 h is achieved, significantly surpassing other approaches in the literature.

Machine learning (ML) methods establish mathematical models to analyze data, revealing the relationships and patterns within them. In recent years, driven by significant advancements in computing power and the development of big data technologies, ML has found extensive applications in the field of catalyst design[1,2]. The application of ML in catalyst design primarily includes: rapidly screening materials with potential catalytic activity through the analysis of vast experimental and computational data, uncovering the intrinsic relationships between catalyst activity and its inherent features, and optimizing theoretical calculation methods[3,4]. These approaches significantly enhance the efficiency and accuracy of catalyst development, particularly when dealing with complex systems and large datasets[5]. The impact of ML in catalyst research is underscored by notable achievements. Specifically, a practical ML approach involves constructing descriptors for target materials based on existing material properties and subsequently discerning latent relationships between these descriptors and specific reaction performance metrics, such as adsorption energy, energy barriers, turnover frequency (TOF), etc[6,7]. For instance, Tran et al. harnessed ML to establish the correlation between 1,499 distinct intermetallic combinations and their adsorption energies for CO and H, leading to the discovery of highly active electrocatalysts for $CO_2$ reduction and $H_2$ evolution[8].

ML models enable researchers to swiftly identify the most promising candidates from thousands of potential catalysts, a feat nearly impossible with traditional methods. In this work, we employ ML methods to accelerate the development of catalysts for methane cracking.

Natural gas is abundant and could be converted into hydrogen, playing a significant role in human society[9,10]. Among various routes of methane conversion, the direct cracking of methane has garnered more and more attention due to its advantages, including high energy conversion efficiency, easy separation of products, and environmental friendliness[9,11,12]. The typical issue of this reaction is that the solid carbon product gradually accumulates on the metallic active sites both inside and outside the catalyst pores, leading to catalyst deactivation and hindering the stability of the methane cracking reaction[10,13]. Additionally, continuously exploring new catalysis approaches has always been a pursuit of researchers. Thus, addressing carbon deposition to extend catalyst life and further improving catalyst activity are two major challenges for the direct cracking of methane. To address these challenges, extensive research efforts have been dedicated to catalyst design aimed at achieving both high activity and long lifetime[11,14–16]. D. Chester Upham et al. presented a novel technology for producing pure hydrogen and separable carbon by

[1]Institute of Frontier Chemistry, School of Chemistry and Chemical Engineering, Shandong University, Binhai Road No.72, 266237 Qingdao, China. [2]These authors contributed equally: Jikai Sun, Rui Tu. ✉e-mail: yutie@sdu.edu.cn; dengwq@sdu.edu.cn

introducing $CH_4$ into molten metal alloy catalysts[17]. A 27% Ni–73% Bi alloy achieved 95% $CH_4$ conversion and nearly 100% $H_2$ selectivity at 1065 °C in a 1.1-meter bubble column, as reported in the piloting process. Besides, Chen et al. presented a ternary Ni-Mo-Bi liquid alloy by adding Mo into the Ni-Bi alloy, which further improved the hydrogen yield to 0.022 $gH_2$ $gNi^{-1}$ $h^{-1}$ at 800 °C[18].

Herein, we propose the utilization of single-atom alloy catalysts (SAAs) in combination with a ball milling strategy to address the aforementioned concerns of high activity and coke deposition. As previously mentioned, alloy materials among the various $CH_4$ cracking catalysts documented in the literature provide multiple adsorption sites on their active centers, which favorably bind and stabilize dissociation intermediates, resulting in high methane cracking activity[19]. Furthermore, single-atom catalysts (SACs) offer uniform active sites and high selectivity. SAAs effectively merge the advantages of SACs and alloy structures, positioning them as potential candidates for $CH_4$ cracking catalysts[19,20]. The manipulation of SAAs' performance can be achieved through the design of metal surface configurations and the variation of single-atom metal compositions, offering an avenue for catalyst development with high catalytic activity tailored to specific reactions[21,22]. However, the compositional space of SAAs is exceedingly extensive, and screening optimal compositions solely through experimental means proves to be time-consuming and labor-intensive. As mentioned above, ML has emerged to accelerate the screening of optimal compositions. Therefore, we utilize ML to search through the vast compositional space of SAAs, thereby accelerating the development of catalysts.

To tackle the issue of coke removal, the introduction of a ball milling approach through a mechanical vibrations pattern into the $CH_4$ cracking process facilitates the timely separation of deposited carbon powder from the catalyst surface. This inspiration originates from the friction or shear between milling balls under dynamic reaction conditions, facilitating the elimination of deposited coke. Prior to this, mechanochemistry has achieved significant advancements in ammonia synthesis, reporting a novel $NH_3$ production method via $N_2$ hydrogenation and modulation of reaction temperature/pressure among other factors[23–26]. Mechanochemical methods can activate reactants under mild conditions by generating high-density structural defects on catalysts or reaction substrates[27], increasing specific surface area[28], adjusting electronic structure[29], or changing reaction paths[26], thus reducing the temperature and pressure required for the reaction.

In this work, a library, including 10,950 SAA surfaces is constructed by substituting a single host metal atom in the top surface layer with a dopant atom. Then, an ML model has been developed to predict the C-H dissociation energy barriers across all fabricated SAAs surfaces. Based on the insights gained from ML outcomes, the relationship between methane cracking activity and SAAs composition is further established. The SAAs highlighted by the ML model as having high activity potential are subjected to experimental validation, confirming that Ir/Ni and Re/Ni exhibited remarkable activity and selectivity for direct methane cracking. Using the mechanical force contribution stemming from ball milling facilitates the removal of coke from the catalyst surface, thereby extending the catalyst's operational lifespan beyond 200 h.

## Results

### Single-atom alloy catalysts screening through machine learning method

The dehydrogenation of $CH_x$ species (x = 1∼4) is conceptualized as occurring through successive steps, wherein each step entails the removal of a single hydrogen atom. Several studies concur that the dissociation barrier of CH ranks highest among the four-step dehydrogenation processes of $CH_x$[30,31]. Furthermore, as illustrated in Supplementary Fig. 1, the energy barriers for the first and fourth steps of $CH_4$ dehydrogenation are the most elevated and positively correlated.

In light of these insights, a series of SAAs was devised, and the energy barrier for C-H bond decomposition was computed via DFT. The structures and adsorption sites of distinct SAAs surfaces are depicted in Supplementary Fig. 2, yielding a dataset comprising 623 DFT energy barrier data points. Subsequently, an automated workflow for retrieving SAA surface information was established, culminating in a database, including 10,950 entries. Thereafter, two ML classification models were employed to categorize the dataset, followed by an ML regression model to forecast their C-H dissociation energy barriers. Additionally, through the combination of transition state theory (TST) with the proportions of different surfaces on the host metal, the total rate of C-H dissociation was calculated across all SAAs surfaces. Further details regarding the ML workflow and computational methods are available in the Surface model and Machine learning sections in the supplementary information, including Supplementary Figs. 4–13 and Supplementary Tables 1–4.

The outcomes of ML predictions are depicted in Fig. 1 and also in Supplementary Fig. 13. In Fig. 1a, the collective C-H dissociation rates are displayed for all formulated SAA surfaces at 450 °C. Notably, SAAs with host metals like Fe, Co, and Ni demonstrate commendable C-H dissociation activity, in line with prior experimental findings[10,13,32–34]. Moreover, single metal atoms loaded onto these host metals, including Co, Ru, Re, Os, and Ir, exhibit the highest C-H dissociation rates, positioning them as potential candidates for optimal catalysts in methane cracking. Additionally, an exploration of the relationship between ML descriptors and C-H dissociation barriers was undertaken to elucidate the mechanisms dictating SAA activity. The analysis of feature importance for all employed descriptors is presented in Fig. 1b, with the most significant descriptor being doped_weighted_surface_energy, accounting for over 40% of the overall feature importance. This particular descriptor pertains exclusively to doped single-atom metals. Other highly ranked properties include com_top_d_e_number, host_molar_volume, com_top_d-band, CN-B3 + 1-top05, com_top_electronegativity, host_surface_energy, and host_surface_work_function. As depicted in Supplementary Fig. 5, the Pearson correlation coefficients among com_top_d_e_number, com_top_d-band, CN-B3 + 1-top05, host_surface_energy, and host_surface_work_function are notably elevated. These descriptors predominantly capture surface coordination numbers and d-electron attributes of the SAAs. Furthermore, host_molar_volume indicates the atomic radius size of the host metal, while com_top_electronegativity conveys electronegativity details of both single-atom metals and host metal pairs. The distribution and correlation between C–H dissociation activities and these descriptors are illustrated in Fig. 1c and Supplementary Fig. 12. Generally, the energy barrier generally declines with an increase in doped weighted surface energy. For descriptors such as d-electron number and electronegativity, the energy barrier initially declines before ascending. Figure 1c offers insight into the distribution of SAAs with doped_weighted_surface_energy and com_top_d_e_number, indicating the existence of two distinct high-activity regions. These regions are characterized by doped_weighted_surface_energy values ranging from 2.5 to 3.3, primarily corresponding to elements like Ir, Tc, Ru, Mo, Re, and others. Similarly, com_top_d_e_number falls within the ranges of 20−30 and 35−45, primarily aligned with eighth subgroup elements featuring low coordination numbers.

### Catalytic performance of $CH_4$ cracking under mechanical reaction conditions

Based on the ML prediction results, the optimal SAAs were composed of Co, Ru, Re, Os, Ir, etc. loaded on Ni or Fe substrates. Subsequently, we conducted experimental tests to investigate the feasibility of synthesizing SAAs and evaluate their activity. Besides serving as substrates, the metal substrates were also employed as milling balls under dynamic vibration conditions. The active metal species predicted to be effective were anticipated to be deposited onto the surfaces of Ni or Fe

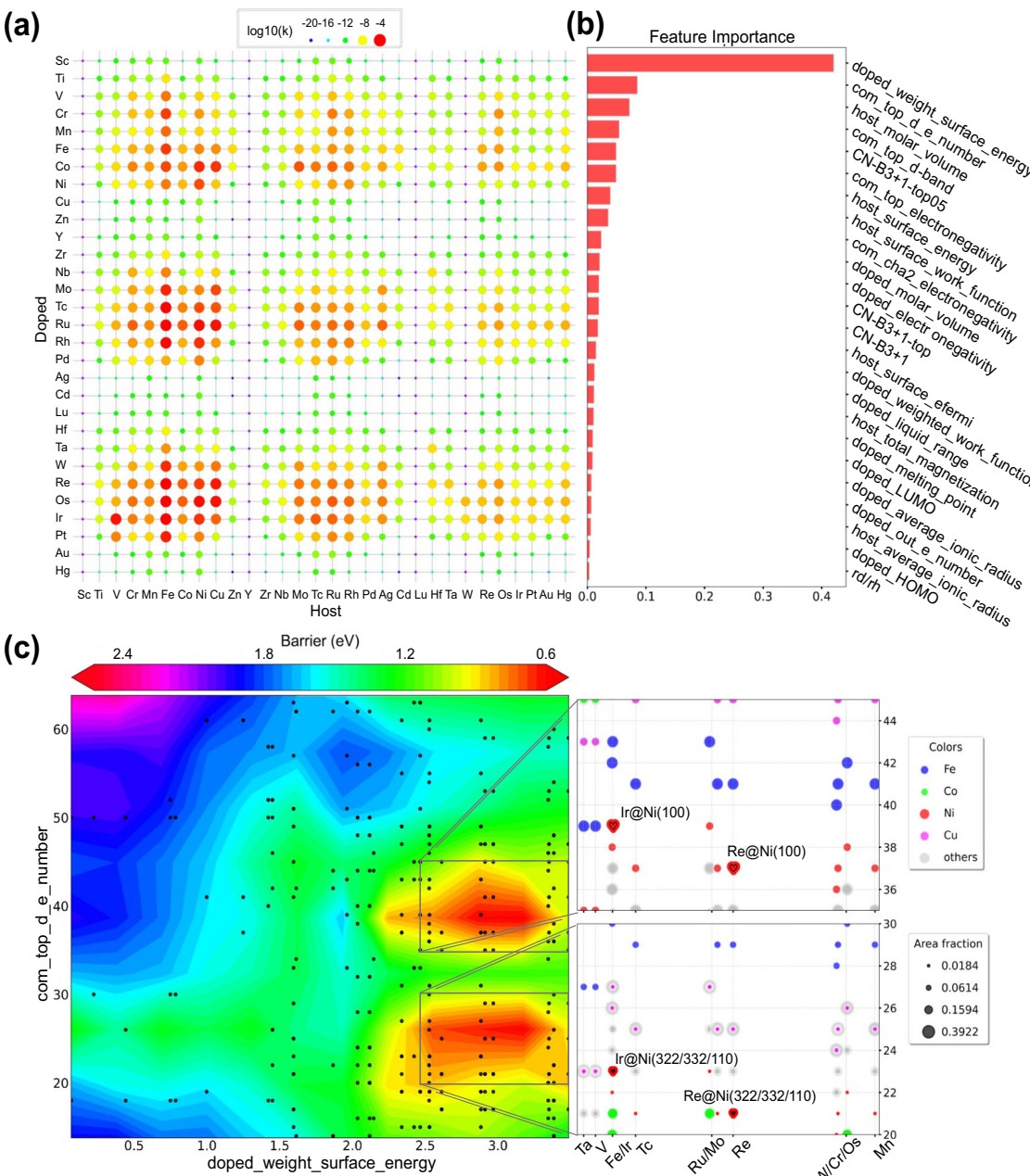

**Fig. 1 | Single atom alloy catalyst designed by machine learning model. a** Total C-H dissociation rate on all surfaces of SAAs at 450 °C. **b** Ranking of feature importance for 24 descriptors. **c** Two-dimensional volcano plot illustrating C-H dissociation energy barrier with respect to scattered SAA surfaces.

balls. The synthesis of SAA balls involved employing a solution impregnation method followed by high-temperature reduction under $H_2$ atmosphere. Additionally, to remove coke and enhance catalyst longevity, a ball milling approach was introduced. This method harnessed mechanical friction and shear generated by the collision between milling balls during vibration. The assessment of methane cracking performance was performed under atmospheric pressure, and the design of the custom ball milling reactor is presented in Supplementary Fig. 14. For comparison, control experiments were conducted using pure Ni/Ni metal balls prepared via the same procedures as the SAA catalysts. Despite the high reaction activities exhibited by Co, Ru, Re, Os, or Ir-doped Fe balls, our experimental findings indicated that extensive calcination led to the delamination of the surface layer of Fe balls, causing the detachment of the active metal layer. Consequently, the activity of Fe-based SAA balls was compromised (Supplementary Table 5). In contrast, when Ni balls were

employed as the host, their intact surface structure after solution impregnation and high-temperature calcination favored the formation of an alloy structure. Thus, this investigation determined Ni balls as the ideal substrate for immobilizing active metal species. In Fig. 2a, both Ir/ Ni and Re/Ni demonstrated superior $H_2$ production rates and $CH_4$ conversion rates compared to pure Ni balls.

We first studied the effects of reaction temperature and mechanical vibration on the catalytic efficiency of Re/Ni and Ir/Ni. Keeping safety requirements in mind, it is advisable to limit the maximum operating temperature of the mechanical reactor to 450 °C. Within this temperature range, the yield of methane increased with rising temperature due to the endothermic nature of methane cracking reactions in Fig. 2b. In line with ML predictions, Ir/Ni SAA demonstrated higher activity, with a hydrogen yield of 13.3 $gH_2$ gcat$^{-1}$ h$^{-1}$, a selectivity of 99.9%, and $CH_4$ conversion of 13.87% at 450 °C and 1 atm. In addition, compared with the static condition, the methane

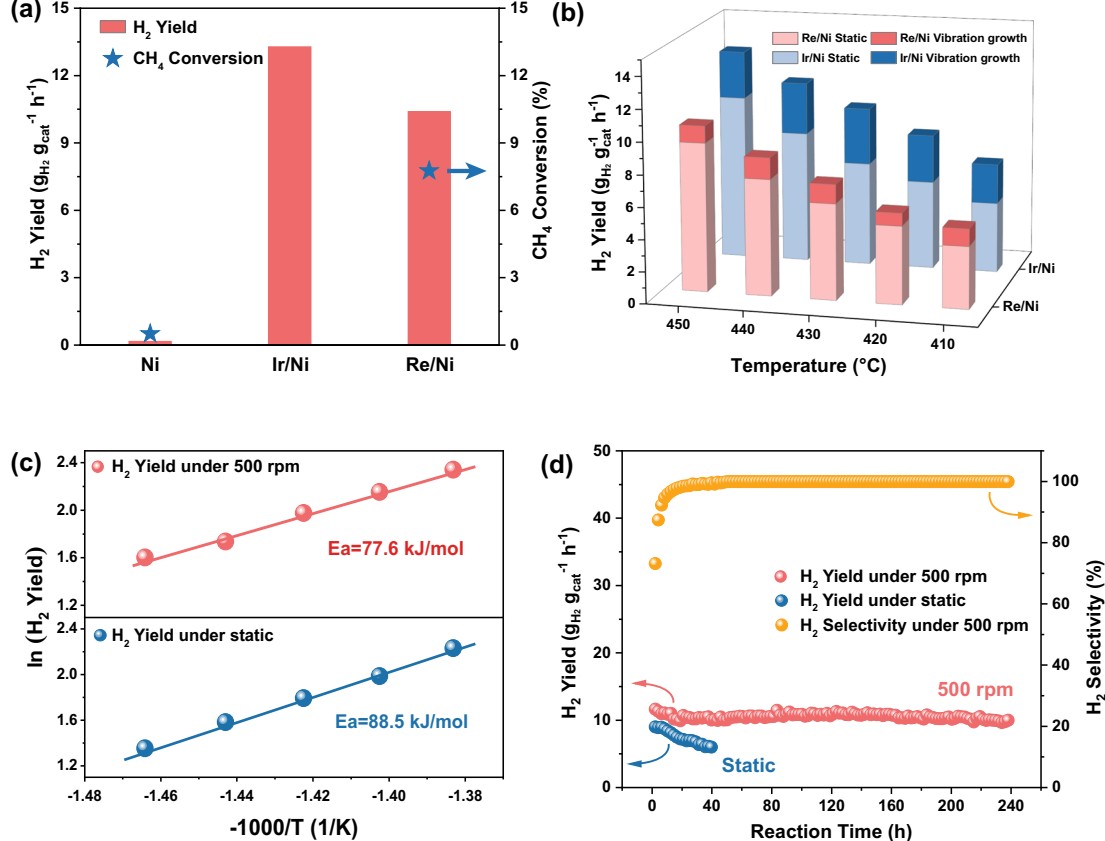

**Fig. 2 | Catalyst performance of CH₄ cracking. a** Comparison of CH₄ conversion and H₂ yield across various catalysts. Ball milling reaction conditions: 50 mL/min CH₄, 450 °C, 500 rpm motor speed. **b** H₂ yield comparison over Re/Ni and Ir/Ni under static and mechanical vibration conditions at temperatures ranging from 410 °C to 450 °C. The dark color part represents the improvement of catalytic effect from mechanical catalysis. Reaction conditions: 50 mL/min CH₄. Ball milling entails a motor vibration frequency of 500 rpm, whereas static catalysis employs 0 rpm vibration frequency. **c** Activation energy comparison of Re/Ni under different reaction conditions. **d** Long-term study of CH₄ decomposition over Re/Ni at 450 °C with a 500 rpm vibration frequency or under static conditions.

yield under mechanical conditions is clearly improved, especially for Ir/Ni, the methane yield under mechanical conditions at 450 °C is 29.1% higher than that under static conditions. Furthermore, the decrease in apparent activation energy (Fig. 2c) observed under mechanical conditions underscored the positive impact of mechanical forces on methane cracking compared with that under static conditions. For the Ir/Ni with the highest activity, the influence of space velocity on the reaction activity was examined. The results listed in Supplementary Fig. 15 showed that the lower the space velocity, the higher the methane conversion. When the methane flow rate was reduced to 5 mL/min, the methane conversion could even reach 27.2%, because the long contact time between catalyst and methane benefited CH₄ conversion under the low gas flow rate.

To comprehend the role of mechanical energy on CH₄ cracking, DFT calculations were performed on Re/Ni and Ir/Ni catalysts. The Ni (111) surface, representing a significant proportion, served as the theoretical model. The connection between the macroscopic collision of Ni balls and the microscopic deformation of the model was established through axial stress resulting from the collision. As shown in Supplementary Fig. 16, the deformed model exhibited compression by 10% in the Z-direction and extensions by 3% in both the X and Y directions. The simulation of C-H bond cleavage was carried out on the deformed surfaces of Re/Ni (111). DFT calculations indicated that the energy barrier on the deformed Re/Ni (111) surfaces was 1.17 eV, which was 0.15 eV lower than that on the undeformed surface. The transition state geometries of CH dissociation on the Re/Ni (111) surface suggested that H could migrate from C to the top of the Re atom. This suggests that enhancing the adsorption capability of active metal atoms could

facilitate H dissociation from C-H bonds and enhance its migration to active metals, promoting H₂ generation. The calculated changes in the projected density of states (PDOS) before and after collision-induced deformation were presented in Supplementary Fig. 17. Collision-induced deformation led to a 0.13 eV upward shift in the $d$-band center ($\varepsilon_d$) for Re, signifying an increased adsorption capacity of H on deformed Re/Ni surfaces. Similar alterations were observed on Ir/Ni (111) surfaces. After deformation, the energy barrier for C-H dissociation decreased by 0.14 eV, and the $\varepsilon_d$ shifted upwards by 0.17 eV. As a result, it is anticipated that collision-induced $\varepsilon_d$ upshifts for Ir and Re enhance hydrogen capture capacity and reduce the energy barrier for C-H bond cleavage.

Catalyst stability is a crucial aspect of methane cracking reactions. From a cost perspective, we primarily focused on the stability of the Re/Ni catalyst, given that Re is a non-precious metal. Stability tests were performed at 450 °C with a ball milling motor speed of 500 rpm. As demonstrated in Fig. 2d, Re/Ni exhibited superior stability compared to previously reported results. The H₂ production over Re/Ni decreased from 11.6 gH₂ gcat⁻¹ h⁻¹ to 10.5 gH₂ gcat⁻¹ h⁻¹ within the initial 16 h of the reaction, accompanied by the formation of CO and CO₂ as byproducts. Subsequently, the reaction activity remained stable for an extended duration of 240 h while maintaining nearly perfect H₂ selectivity. In static conditions, similar to traditional fixed-bed reactors, continuous carbon accumulation on catalytic active sites leads to rapid deactivation. Thus, it can be concluded that under mechanical conditions using milling balls as catalysts, the collision and friction during dynamic reactions mitigate carbon buildup and considerably extend the catalyst lifespan. In addition, to comprehend the role of ball

milling in $CH_4$ cracking, calculations of the carbon slip process over the metal surface were conducted in terms of energy barrier and dynamics. The Ni (111) surface, representing a significant proportion, served as the theoretical model. As shown in Supplementary Fig. 18, the CI-NEB calculations presented that the slip energy barrier of the carbon fragment was 0.68 eV and the generated carbon could be easily removed through mechanical vibration in the grinder. Meanwhile, the AIMD simulation in Supplementary Fig. 19 and Supplementary Movie 1 revealed that the carbon fragment could easily slip from the Ni (111) surface after an initial velocity of 0.001 Å/fs was applied. This result demonstrates that the friction resistance of Ni metal surface to carbon sliding is very small. Thus, vibration conditions could effectively eliminate the carbon deposition on the catalyst surface and prolong the catalyst lifetime. Comparing the performance of Ir/Ni and Re/Ni SAAs in this study with previous research, as summarized in Supplementary Table 6, Re/Ni exhibited a combination of high hydrogen yield and prolonged lifespan.

## Structural characterization of catalysts

To determine the micro-environment of active metal species on milling balls, initial X-ray diffraction (XRD) analyses of M/Ni (M = Ir, Re) catalysts were performed. These analyses indicated the presence of Ni and NiO crystalline phases, while M or M-oxide phases were absent (Fig. 3a and Supplementary Fig. 20), suggesting a high dispersion of M species. Furthermore, the presence of NiO phase suggested incomplete reduction during catalyst synthesis. X-ray photoelectron spectroscopy (XPS) data (Supplementary Figs. 21, 22) revealed coexisting $Ni^0$ and $Ni^{2+}$ valence states on M/Ni. The O 1 $s$ XPS signal confirmed NiO presence, potentially linked to CO and $CO_2$ generation during methane cracking. Time of flight secondary ion mass spectrometry (TOF-SIMS) characterized Re distribution on Ni bulk phase. Figure 3b and Supplementary Fig. 23 illustrate Re's uniform distribution within tens of nanometers on Ni's surface, with decreasing content at greater depths (Supplementary Fig. 24c), indicating Re's exclusive presence on the Ni surface. Ir/Ni exhibited similar TOF-SIMS results to Re/Ni (Supplementary Fig. 24). Scanning electron microscopy (SEM) images (Supplementary Fig. 25a, b) and Atomic Force Microscope (AFM) images (Fig. 3c, d) depicted Ni substrate morphology changes pre- and post-Re deposition. Notably, Re deposition induced surface roughness, evident in the significant height difference increase in the AFM image of Re/Ni, which also correlated with the Re signal detected up to 40 nm depth (Supplementary Fig. 23c). Ir/Ni's SEM and AFM characterizations corroborated the exclusive surface doping of Ir atoms (Supplementary Figs. 25c, 26).

To confirm the microstructure of Re/Ni and Ir/Ni at the atomic level, high-angle annular dark-field scanning transmission electron microscopy (HAADF-STEM) was performed (Fig. 3e and Supplementary Fig. 27). The HAADF-STEM image of Re/Ni showed individual Re atoms uniformly distributed over the Ni substrate, and no Re NPs or clusters. Elemental distribution in the corresponding area was also observed by energy dispersive spectroscopy (EDS) mapping images (Fig. 3e). Re is uniformly dispersed on Ni NPs at an atomic level. X-ray absorption near-edge spectroscopy (XANES) and extended X-ray absorption fine structure (EXAFS) were conducted to examine Re species' coordination on Re/Ni catalyst. In Fig. 3f, Re/Ni catalyst's XANES white-line intensity slightly exceeded Re foil but fell below $Re_2O_7$, suggesting $Re^{\delta+}$ species with a positive charge. This $Re^{\delta+}$ species possibly resulted from highly dispersed Re atoms' alloying on Ni substrate, inducing electron transfer from Re to Ni atoms. Re/Ni's EXAFS Fourier transform spectra in R-space displayed a primary peak at 2.3 Å (Fig. 3g), attributed to the Re-Ni alloy structure, distinguishable from Re foil's Re-Re bond or $Re_2O_7$'s Re-O bond. Notably, no Re-Re bond appeared in the EXAFS spectra, signifying the absence of Re particles on Re/Ni. Profile fitting results in R-space indicated a Re coordination number of 7 (Supplementary Table 7), and distinct $k^3$-weighted EXAFS amplitudes in k-space (Supplementary Fig. 28) underscored its SAA structure in Re/Ni compared to Re foil and $Re_2O_7$. Similarly, Ir-Ni SAA structure was confirmed through EXAFS spectra for Ir/Ni (Supplementary Fig. 29 and Supplementary Table 8), highlighting the versatile SAA structure generation on Ni substrate via the employed preparation method for different active metal species.

## The application of solid carbon powder (by-product) as electrode in lithium battery

Following the ball milling reaction, the sole solid-phase product, carbon, is separated from the milling balls, collected, and characterized. The SEM image in Fig. 4a illustrates the filamentous structure of the generated carbon powder. Energy dispersive spectroscopy (EDS)

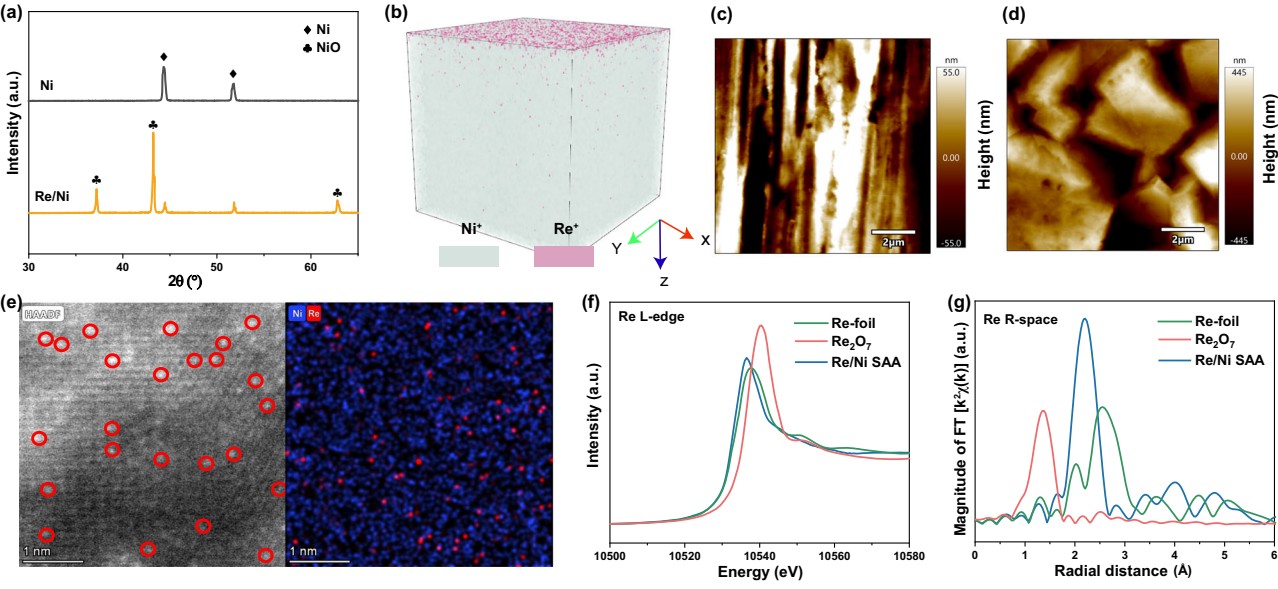

**Fig. 3 | Catalyst characterization. a** XRD profiles of pristine Ni and Re/Ni. **b** 3D TOF-SIMS map of the Re signal on the Ni substrate. The 2D surface geometry images of Ni (**c**) and Re/Ni (**d**) sample from AFM. **e** HAADF-STEM image and elemental mapping of Re/Ni. **f** XANES spectra of Re $L_3$-edge from Re/Ni catalyst with Re foil and $Re_2O_7$ as references. **g** Re $L_3$-edge EXAFS in R space from Re/Ni catalyst with Re foil and $Re_2O_7$ as references.

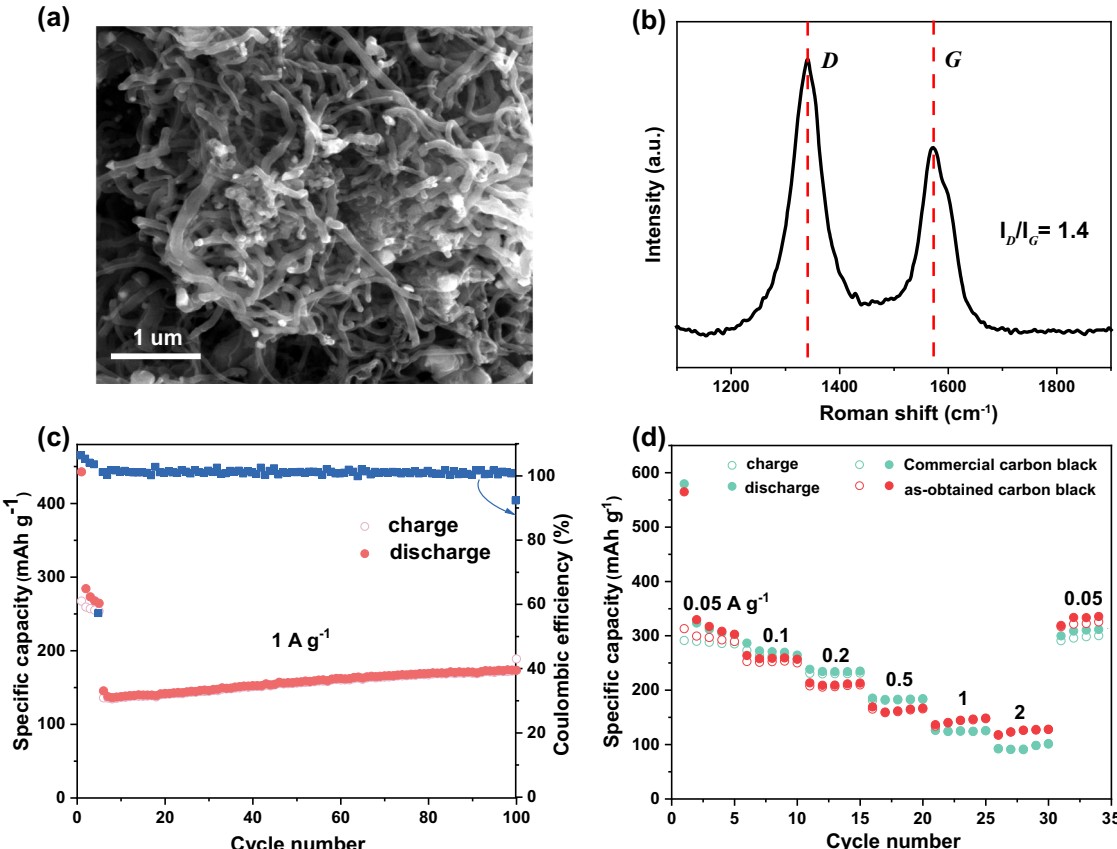

**Fig. 4 | Characterization and recycling of carbon products. a** SEM image of carbon species formed during the reaction process. **b** Raman spectra of carbon species. **c** Cycling stability of the as-synthesized carbon black and commercial carbon black electrodes for current densities ranging from 0.1 to 1 A g⁻¹. **d** The rate performance of as-synthesized carbon black and commercial carbon black electrodes from 0.05 to 2 A g⁻¹.

characterization results (Supplementary Fig. 30) show no detection of Re substance in the carbon powder after the reaction, confirming the intact state of active metal atoms on the Ni ball at this vibration frequency. The Raman spectrum (Fig. 4b) and XRD profile (Supplementary Fig. 31) attribute the collected carbon product to carbon black based on the $I_D/I_G$ ratio. Supplementary Fig. 32 illustrated no $CH_4$ cracking activity over collected carbon products. To enhance carbon black's application potential, it was used as an electrode material to assess its lithium storage capability. Figure 4d presents a comparison of the rate performance between the as-synthesized carbon black and commercial carbon black electrodes at current densities ranging from 0.05 to 2·A·g⁻¹. The as-synthesized carbon black electrode demonstrates superior reversible capacities of 302.7, 256.7, 213.0, 166.9, 148.5, and 128.0 mA·h·g⁻¹ at 0.05, 0.1, 0.2, 0.5, 1 and 2 A·g⁻¹, respectively. Therefore, it can be asserted that their rate performance was much better compared to that of commercial carbon black. In addition, when the current density was returned to 0.05 A·g⁻¹, the as-synthesized carbon black electrode recovered its capacity of 302 mA·h·g⁻¹, which demonstrated its good electrochemical reversibility. Moreover, it exhibited good cycling stability (Fig. 4c), maintaining a reversible capacity of 232.9 mA·h·g⁻¹ after 400 cycles at 1 A·g⁻¹. In comparison to commercial carbon black, the product from methane conversion exhibits superior cycling performance. Since the obtained by-product carbon powder could be utilized as a commercial product such as electrode material in lithium battery and increases the atomic economy of this methane cracking approach.

In order to prove that mechanical catalysis approach for carbon removal during methane cracking process presented the potential of amplification application, we used a common rotating reactor in industry to conduct methane cracking experiments. The schematic diagram of home-made was presented in Supplementary Fig. 33. The activity results showed that the Ir/Ni SAA exhibited stable $H_2$ production in 200 h without deactivation (Supplementary Fig. 34). It was worth noting that the activity was slightly lower than that via vibration reactor, which was induced by weak collision force between milling medium in rotating reactor. While the produced carbon powder could also be removed from catalyst surface in rotating reactor, further reactor design is necessary to regulate the collision strength and gaseous flow route to improve $H_2$ production.

In conclusion, a ML workflow has been devised to predict methane cracking on SAAs. This workflow automatically acquires properties and surface information of SAAs, selects optimal descriptors, and performs ML classification and prediction. Utilizing this method, numerous potential methane-cracking catalysts were screened from 10,950 surfaces of transitional metal SAAs. After materials synthesis and activity testing, Ir/Ni and Re/Ni were validated to possess exceptional catalytic activity for methane cracking. Furthermore, when coupled with the ball milling approach, Re/Ni achieved a remarkable record-breaking lifetime of 240 h for pure $H_2$ production from $CH_4$ cracking. Besides, the by-product carbon could be used in lithium battery and exhibits higher performance than commercial carbon black. This work establishes a paradigm for designing targeted catalysts for specific reactions within vast structural spaces.

## Methods

### Preparation of the M/Ni (M = Re, Ir, Ni) SAAs

Ni balls, weighing 333 g with diameters of 8–10 mm, were procured from Aladdin. Before loading metals, the Ni balls were sonicated in N, N-Dimethylformamide (DMF) for 30 min and washed with deionized

water. After drying, they were immersed in a solution of perrhenic acid (0.0127 M) for 10 h. Subsequently, filtration, air-drying for 30 min, calcination at 850 °C, and reduction at 700 °C were performed under an atmosphere of 20 mL/min 10% $H_2$/Ar. Inductively coupled plasma emission spectrometer (ICP) results confirmed the weight ratio of Re/Ni to be 12 mg/1 kg. For the loading of Ir/Ni SAAs, chloroiridic acid (0.0127 M) was used as a precursor with similar loading procedures. ICP results confirmed the weight ratio of Ir/Ni to be 16.9 mg/1 kg. In order to eliminate the catalytic activity of Ni balls, we prepared Ni/Ni balls by the same method using Ni$(COOH)_2$ as precursor.

### Sample characterization

X-ray diffraction (XRD) patterns were obtained using a PANalytical Model Xpert3 instrument with a Cu Kα radiation source (λ = 0.15406 nm) operating at 40 kV and 10 mA. X-ray photoelectron spectroscopy (XPS) was conducted on a ThermoFisher ESCALAB 250Xi spectrometer using a monochromatized Al Kα X-ray source (1486.6 eV) with an applied power of 150 W. The C $1s$ peak (binding energy 284.8 eV) served as a reference. Ir and Re loading was estimated through inductively coupled plasma–optical emission spectrometry (Agilent 725 ICP–OES). SEM investigations utilized a Quanta 250 FEG instrument to analyze carbon product morphology. Raman spectra of carbon products were acquired with a Thermo Fisher D XR2 spectrometer using an excitation wavelength of 532 nm and a laser power of 8 mW. Prior to measurement, the Raman spectrometer was calibrated with the silicon peak (520.7 cm$^{-1}$). A thermal infrared imager (AVIO NEC R450) was employed to determine Ni ball temperatures before and after collision, while atomic force microscopy (AFM, SPI 3800 N SPA400) was utilized for metal surface morphology measurements. Time-of-flight secondary ion mass spectrometry (TOF-SIMS) full spectra are used to study Re/Ir spatial localization with lateral resolution to the sub 50 nm region. The depth profiling was performed on a TOF-SIMS 5-100 instrument (ION-TOF) GmbH, Müenster, Germany) using a 30-KeV $Bi_3^+$ as analysis beam and 1-keV $Cs^+$ ($O_2^+$) as sputter source for positive (negative) polarity measurements. The analysis beam current was 0.7 pA, and the raster size was $50 \times 50\ \mu m^2$. The X-ray absorption spectra (XAS) of Re/Ir L-edge are collected at the beamline of TPS44A1 in the National Synchritrin Raduatuib Research Center (NSRRC), Taiwan. The spectra were recorded under room temperature with fluorescence mode with a solid-state detector. The focused ion beam (FIB) technique was utilized to prepare the metallic slice for High Angle Angular Dark Field-Scanning Transmission Electron Microscopy (HAADF-STEM) imaging. The morphology of Re/Ni catalysts was characterized on an aberration-corrected HAADF-STEM instrument at 300 kV (FEI Themis Z). The energy-dispersive spectrometer (EDS) mappings were performed on the FEI Themis Z microscope.

### Computational details

First-principles calculations were conducted using the Vienna ab initio Simulation Package (VASP)[35,36]. The projector increased wave (PAW)[37,38] potentials were employed for electron-ion interaction and the generalized gradient approximation (GGA) method with the Perdew, Burke, and Ernzerhof (PBE)[39] functional was used for exchange and correlation potential[40]. The DFT-D3 method proposed by Grimme[41] was incorporated to account for intermolecular van der Waals (vdW) interactions. An energy cut-off of 400 eV ensured accuracy. For surface calculations, electronic self-consistency was set with a tolerance of $1 \times 10^{-5}$ eV, and Hellmann–Feynman forces on free atoms during ionic relaxation were optimized to be under $2 \times 10^{-2}$ eV Å$^{-1}$. The Monkhorst-Pack method sampled the Brillouin zone, ensuring k × a > 30[42]. The climbing image nudged elastic band (CI-NEB) method and dimer method were used for locating transition states (TS), and stretching frequencies were analyzed to identify a TS with only one imaginary frequency[43,44].

The carbon slip process on Ni (111) surface was performed using an ab initio molecular dynamics (AIMD) simulation on the CP2K package[45]. To minimize basis set superposition errors, the wave functions were expanded in the DZVP-MOLOPT-SR-GTH basis set[46,47] along with the cutoff energy of 500 Ry for the auxiliary plane-wave basis set. Core electrons were modeled by Geodecker–Teter–Hutter (GTH) pseudopotentials[48–50]. Only the Γ-point approximation was employed to sample the Brillouin zone[42]. First, the AIMD simulations were equilibrated in the canonical ensemble (NVT) with the Nose–Hoover thermostat[51,52] at 723.15 K for 800 fs of equilibrated trajectory with a time step of 1.0 fs. Then, an initial velocity of 0.001 Å/fs was applied to the carbon fragments and further simulated for 1000 fs.

## Data availability
Source data are provided with this paper. The data that support the findings of this study are also available from the corresponding author [T. Y. and W. Q. D.] upon reasonable request. Source data are provided with this paper.

## Code availability
The code used to construct metal surfaces and obtain descriptors is available at https://zenodo.org/record/8133294.

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

## Acknowledgements

This work was supported by the National Key R&D Program of China (No. 2022YFA1503104 to W.D.), Taishan Scholars Project (No. tspd20230601 to W.D.), the Fundamental Research Funds of Shandong University (No. 2019GN111 to T.Y.), State Key Laboratory of Catalysis funds of Dalian Institute of Chemical Physics (No. N-22-16 to T.Y.), the Natural Science Foundation of Shandong Province (No. ZR2023QB204 to T.Y.), and the Shandong University Future Program for Young Scholars (No. 62460082064083 to T.Y.). The authors are grateful for the technical support for Nano-X from Suzhou Institute of Nano-Tech and Nano-Bionics, Chinese Academy of Sciences (SINANO). The authors thank Xiaoju Li and Haiyan Sui from Shandong University Core Facilities for Life and Environmental Sciences for their help with the TEM. Thanks to Hefei Kejing Material Technology Co., Ltd. for its support for reactor manufacture. We thank Honglei Wang, Guoqing Ren, Chengcheng Liu, Li Yang, Lei Sun, and Zhen Li at Shandong University for their input to the manuscript.

## Author contributions

W.D. supervised the project and conceived the idea. J.S. performed the DFT and machine learning studies. J.S. and D.Z. discussed the machine learning results. R.T., Y.X., X.C., and T.Y. designed and carried out the catalysis experiments. H.Y. carried out the battery test. All authors discussed the results and assisted during manuscript preparation.

## Competing interests

The authors declare no competing interests.
