## [Peer Review File · Nature Communications]

Machine Learning Aided Design of Single-atom Alloy Catalysts for Methane CrackingReviewers' comments:

Reviewer #1 (Remarks to the Author):

The submitted manuscript from 14 authors reports to use machine learning to identify single atom alloy catalysts (SAAC) to use for methane pyrolysis and claims to “pioneer a direction for economic hydrogen production”

The paper presents some potentially very interesting and novel applications of machine learning to catalysis research. The authors combine first-principles calculations with a compressed-sensing data-analytics approach, utilizing SISSO which allowed for the prediction of catalytic properties across a large number of SAAC candidates. The study claims to have identified over 200 previously unreported SAAC candidates which may have improved performance at very low temperature and atmospheric pressure in the very unusual ball milling reactor environment used in the study. The paper may contribute to the field by introducing advanced computational techniques for the design and analysis of SAAC's possibly facilitating the discovery of new catalysts. The authors identified Ir/Ni and Re/Ni as potential catalysts for methane pyrolysis, however, no evidence is provided that the catalysts experimentally measured were single atoms.

The paper might be very inspirational to the catalysis community as among the first demonstrations of new catalyst discovery from searches employing machine learning. Unfortunately, as submitted the paper can not be accepted as it has no value in the area of hydrogen production from methane and it is misleading in all aspects related to commercial hydrogen production. The authors seem unaware that methane pyrolysis is a reversible reaction and that an economical process can not operate at one atmosphere pressure with the low (near equilibrium) conversion demonstrated by the authors (~8% conversion) even with a traditional reactor. It is naïve to think the ball mill reactor proposed by the authors is a practical design for commercial hydrogen production. It is not. The authors propose this reactor to physically dislodge the carbon from the catalyst and attrit off the surface layer of carbon to recover the catalyst activity. The same basic approach has been previously suggested using much more practical fluidized bed reactors. The selection of Ir/Ni or Re/Ni suggests a lack of understanding of the carbon formation process on metal catalysts and the likelihood that significant amounts of Ir or Re will be attached to the carbon leaving the reactor which will never be cost-effective to lose. Further, how do these catalysts even relate to “Single Atom Alloy Catalysts” clearly the catalyst consists of far more than single atoms and we are left guessing as to what the actual catalysts are.

Although the paper's application of machine learning is interesting, the authors should be encouraged to, 1) remove all mention of commercial hydrogen production, 2) focus the paper on identification of new mixed metal catalysts (not single atom catalysts), and 3) resubmit the paper to a more specialized catalysis journal. This is not suitable for Nature Communications.

Reviewer #2 (Remarks to the Author):

The manuscript “Accelerated Design of Single-atom Alloy Catalysts for CO₂-free Hydrogen Production from Methane Pyrolysis Aided by Machine Learning” describes the multidisciplinary work in the area of hydrogen production through thermic methane cracking.

The work includes first modeling from first-principles methane dissociation on a set of single-atom alloy catalysts, followed by the training of AI models and screening of above ten thousands of candidates. Next the authors identified in this way most promising candidates with lowest dissociation barriers, synthesized some of them and tested experimentally. Finally based on experiments the authors built an apparatus for hydrogen production with ball milling technology. The work represents the state-of-the-art of heterogeneous catalysis, and thus can be published in Nature Communications journal.

Before publishing, however, some points need to be addressed:

- it is not very clear why the authors used only recall metric for classification, and not the accuracy in addition? Also the results of classification tests are missing, in contrast to regression as in Figures S8-S10, this should be also shown.
- it is not very clear what was the reason for features preselection step although discussed on P.3 in SI. Most of used ML methods can handle all 84 features with no significant increase of fitting time. Especially non-linear methods should be able to tackle the intercorrelation problem. It makes sense to do such preselection for linear regression, ridge-regression and LASSO, but regarding other methods if prediction accuracy of models containing only preselected features is better than for models obtained with the same ML method but with all 84 features, this should be shown. Besides that, the RFECV method requires an estimator (ML method) which in turn requires tuning the hyperparameters, meaning that final result of such preselection step might depend on this choice.
- the overall impression about ML part is that the authors tried all methods they know without wise understanding why it is worth or worthless to use each method. It is a bit surprising that for relatively small training sets (182, 88 and 353 samples) the authors applied ML methods designed for thousands of data points like neural nets, XGBoost. Although the results of tests (at least for regression, for classification nothing is presented) justify used methods, but their initial choice needs to be discussed, as well as their performance.
- 'doped_weighted_surface_energy' – how was the surface formation energy calculated keeping in mind that one side of slabs was always frozen?
- P.8. The authors write that small observed activity of Fe-based SAA's was caused by the detachment of the active metal layer. Have they calculated stability of corresponding SAA catalysts in terms of aggregation of atoms into particles and segregation? Probably this could immediately indicate that catalysts are unstable.
- in Figure S1 the best found catalyst Re/Ni is not shown, probably it would be worth to show the path for it as well.
- the authors adopted ball milling reactor which performs the up-down vibrational movements. What about a rotating reactor as an alternative? Can it be technically more easily implemented, and would it be more or less efficient in coke separation?

Minor suggestions:

- since the authors show the industrial application of developed catalysts, it would be worth to estimate the overall economic expenses, profitability of such production of hydrogen and carbon shown in Fig. 4a.
- SI P. 7. "The average values of r^2 , RMSE, and MAE for GBDT in the 1000 trials on the test set were 0.913, 0.136, and 0.010 eV" – probably last number is 0.100 eV.

Reviewer #3 (Remarks to the Author):

Jikai Sun et al. describes the study in using machine learning-predicted single-atom alloy catalysts (SAAs) in combination with a ball milling reactor to study methane pyrolysis.

Although this manuscript widely covers a discussion among the machine learning for catalyst scoping, catalyst synthesis, characterization and test, ball mill reaction system, hydrogen separation, as well as carbon products characterization and application, there are many gaps and details without addressing. Therefore, as submitted, it can not be accepted by Nature Communications for publication. Below are the comments from this reviewer.

1) The paper motivates pyrolysis for hydrogen production as requiring lower temperatures and thus more active catalysts. This is a naïve statement since commercially pyrolysis will require high single pass conversion (>~70%) at pressures above approximately 5 -10 bar, thermodynamics will set lower limits on temperature that are ignored in this paper. 7.75 % single pass conversion at near atmospheric conditions does not have much relevance for commercial scale hydrogen production and thus any extrapolation to commercial hydrogen production not supported.

2) In looking over that prior work the authors of this manuscript incompletely interpret reported results about the reaction temperature. The manuscript compares this work (solid catalyst) to a previous paper (liquid catalyst) by Upham et. al, published in Science, to show the advantages of low temperature pyrolysis in this study (1000oC Vs 450). It is comparing apple to orange. Methane pyrolysis with solid catalyst was reported could carried out over low temperatures, for example, with Ni-Cu-Co catalyst, methane pyrolysis could achieve 15 % single pass conversion under 500 oC [A.C. Lua, H.Y. Wang / Applied Catalysis B: Environmental 156–157 (2014) 84–93], which is higher than the results in this manuscript. And in the Ref 12 cited in this manuscript, with a similar liquid metal alloy based catalyst as Upham et. al, the reaction temperature can reach as low as 450 oC.

3) Furthermore, in the introduction section, “Generally, although the above works successfully solve coke removal from catalyst bed, the energy consumption of CH₄ cracking is still significantly enormous.” The authors should evaluate the energy consumption of ball mill reaction system with bubbler reaction system. And especially, under a low methane single pass conversion, the energy consumption of membrane separation system, and methane recycling system compared to a high methane conversion of 70 %.

Therefore, a 7.75 % single pass conversion under low temperature of 450 oC, with solid catalyst did not bring any scientific breakthrough in methane pyrolysis. And the authors should revise their discussion in abstract and introduction section.

4) The authors report the SAA catalyst Re/Ni achieves high hydrogen yield of 10.7 gH₂ gcat h⁻¹ with a calculation of catalyst mass only containing single Re metal. Since the weight ratio of Re/Ni is 12 mg/1 kg, the high yield is meaningless. And how the hydrogen yield of pure Ni sample in Figure 2 a was calculated? And to compare the activity of Ni with Re, the authors need to show the methane conversion data of Ni/Ni sample using the similar synthesis to load Ni on Ni ball, since in Figure 3 C and D, before and after loading Re, the catalyst surface (especially surface area) are different. And the high activity of Re/Ni sample could be originated from the difference of surface area. BTW, the E and F in figure 3 are not in the same scale bar.

5) The authors claim that the Re is in the form of a single atom on the Ni catalyst surface. The authors did not provide strong evidence to support that conclusion. And the authors should also provide more information about the spent catalyst to see if there is Re sintering or sinking on and into Ni substrate. By the way, the carbon atom can be soluble in Ni, does carbon solubility affect the reaction kinetics? And if Re doping affects carbon solubility? Is there any metal carbide formation?

Therefore, the authors should provide more details of their catalyst, and revise their evaluation and discussion in the “catalyst” section.

6) As shown in Figure 2 d, the enhancement of the catalyst stability was due to the ball mill which could remove the carbon from catalyst surface. But that do not exclude the effect of carbon nanotube on methane conversion, since carbon nanotube also displays activity and

used as catalyst previously.

7) The authors should provide more information on catalyst synthesis, engineering parameters of the reaction system and membrane separation system.

8) The authors should provide their vision of reaction system scale up.

Response to reviewers' comments

Reviewer #1 (Remarks to the Author):

The submitted manuscript from 14 authors reports to use machine learning to identify single atom alloy catalysts (SAAC) to use for methane pyrolysis and claims to “pioneer a direction for economic hydrogen production”.

The paper presents some potentially very interesting and novel applications of machine learning to catalysis research. The authors combine first-principles calculations with a compressed-sensing data-analytics approach, utilizing SISO which allowed for the prediction of catalytic properties across a large number of SAAC candidates. The study claims to have identified over 200 previously unreported SAAC candidates which may have improved performance at very low temperature and atmospheric pressure in the very unusual ball milling reactor environment used in the study. The paper may contribute to the field by introducing advanced computational techniques for the design and analysis of SAAC's possibly facilitating the discovery of new catalysts. The authors identified Ir/Ni and Re/Ni as potential catalysts for methane pyrolysis, however, no evidence is provided that the catalysts experimentally measured were single atoms.

The paper might be very inspirational to the catalysis community as among the first demonstrations of new catalyst discovery from searches employing machine learning.

Unfortunately, as submitted the paper can not be accepted as it has no value in the area of hydrogen production from methane and it is misleading in all aspects related

to commercial hydrogen production. The authors seem unaware that methane pyrolysis is a reversible reaction and that an economical process can not operate at one atmosphere pressure with the low (near equilibrium) conversion demonstrated by the authors (~8% conversion) even with a traditional reactor. It is naïve to think the ball mill reactor proposed by the authors is a practical design for commercial hydrogen production. It is not. The authors propose this reactor to physically dislodge the carbon from the catalyst and attrit off the surface layer of carbon to recover the catalyst activity. The same basic approach has been previously suggested using much more practical fluidized bed reactors. The selection of Ir/Ni or Re/Ni suggests a lack of understanding of the carbon formation process on metal catalysts and the likelihood that significant amounts of Ir or Re will be attached to the carbon leaving the reactor which will never be cost-effective to lose. Further, how do these catalysts even relate to “Single Atom Alloy Catalysts” clearly the catalyst consists of far more than single atoms and we are left guessing as to what the actual catalysts are.

Although the paper’s application of machine learning is interesting, the authors should be encouraged to, 1) remove all mention of commercial hydrogen production, 2) focus the paper on identification of new mixed metal catalysts (not single atom catalysts), and 3) resubmit the paper to a more specialized catalysis journal. This is not suitable for Nature Communications.

Response: Thank you for your review and suggestions. We appreciate the opportunity to clarify certain aspects of our work. Regarding the concern about commercial hydrogen production via mechanical catalysis approach in this study, we

have removed those comments about commercial pure hydrogen production and Pd membrane separation system via mechanical catalysis approach already. Instead, we believe that the outlets gas flow via mechanical catalysis approach, with a methane content of approximately 80-85%, could potentially be used directly as hydrogen-enriched natural gas.

In response to the reviewer's comments regarding fluidized bed reactors for H₂ production from CH₄ cracking, it should be noted that these two approaches are completely different from reaction mechanism. For carbon tube formation and H₂ production from CH₄ cracking in fluidized bed reactors, the active metal particles would be sealed inside the carbon tube and leave from reactor with carbon products finally. Nevertheless, for carbon fiber formation and H₂ production via mechanical CH₄ cracking in vibration reactor, the generated carbon fiber was solid without hollow structure, and the active Re and Ir sites were stable on the milling ball surface. It was confirmed that the solid carbon fiber didn't contain the active metal species from composition analysis results. More importantly, mechanical collision between milling balls could further boost CH₄ cracking efficiency due to the induced deformations, which was also different with fluidized bed reactors. We have supplemented our manuscript with relevant calculations to support the boosting impact of mechanical catalysis on Page 9, Figure S16 and Figure S17 as follow.

“To comprehend the role of mechanical energy on CH₄ cracking, DFT calculations were performed on Re/Ni and Ir/Ni catalysts. The Ni (111) surface, representing a significant proportion, served as the theoretical model. The connection

between the macroscopic collision of Ni balls and the microscopic deformation of the model was established through axial stress resulting from the collision. The deformed model exhibited compression by 10% in the Z-direction and extensions by 3% in both the X and Y directions. The simulation of C-H bond cleavage was carried out on the deformed surfaces of Re/Ni (111). DFT calculations indicated that the energy barrier on the deformed Re/Ni (111) surfaces was 1.17 eV, which was 0.15 eV lower than that on the undeformed surface. The transition state geometries of CH dissociation on the Re/Ni (111) surface suggested that H could migrate from C to the top of the Re atom. This suggests that enhancing the adsorption capability of active metal atoms could facilitate H dissociation from C-H bonds and enhance its migration to active metals, promoting H₂ generation. The calculated changes in the projected density of states (PDOS) before and after collision-induced deformation were presented in Figure S17. Collision-induced deformation led to a 0.13 eV upward shift in the d-band center (ϵ_d) for Re, signifying an increased adsorption capacity of H on deformed Re/Ni surfaces. Similar alterations were observed on Ir/Ni (111) surfaces. After deformation, the energy barrier for C-H dissociation decreased by 0.14 eV, and the ϵ_d shifted upwards by 0.17 eV. As a result, it is anticipated that collision-induced ϵ_d upshifts for Ir and Re enhance hydrogen capture capacity and reduce the energy barrier for C-H bond cleavage.”

Fig. S16 Collision deformation model of the Ni (111) surface.

Fig. S17 a, The projected density of states (PDOS) of Ir in Ir/Ni(111) before (top) and after (bottom) collision deformation. b, The PDOS of Re in Re/Ni(111) before (top) and after (bottom) collision deformation.

The experiment also proved that the addition of mechanical force could improve the conversion rate of CH_4 cracking, and the energy consumption for mechanical

vibration only accounts for ~2% of the total energy consumption under ball milling reaction conditions. We supplemented the methane conversion rate of Ir/Ni at different vibration frequencies to prove the boosting influence of mechanical catalysis. Relevant descriptors were added on Page 8, and Fig. 2b as follow.

Fig. 2b H₂ yield comparison over Re/Ni and Ir/Ni under static and mechanical vibration conditions at temperatures ranging from 410 °C to 450 °C. The dark color part represents the improvement of reaction rates induced by mechanical catalysis. Reaction conditions: 50 mL/min CH₄. Ball milling entails a motor vibration frequency of 500 rpm, whereas static catalysis employs 0 rpm vibration frequency.

“Keeping safety requirements in mind, it is advisable to limit the maximum operating temperature of the mechanical reactor to 450 °C. Within this temperature range, the yield of methane increased with rising temperature due to the endothermic nature of methane cracking reactions in Figure 2b. In line with ML predictions, Ir/Ni SAA demonstrated higher activity, with a hydrogen yield of 13.3 gH₂ gcat⁻¹ h⁻¹, a selectivity of 99.9%, and CH₄ conversion of 13.87% at 450 °C and 1 atm. In addition, compared with the static condition, the methane yield under mechanical condition is

improved, especially for Ir/Ni, the methane conversion under mechanical condition at 450 °C is 29.1% higher than that under static condition.”

We have fully considered the possible loss of active metals caused by the vibration of the milling balls. We tested the carbon powder produced after 240 hours reaction over Re/Ni catalyst. At least under the given working conditions (500 rpm), our experimental characterization has confirmed that the carbon powder did not contain active metal Re. To identify the coordination and intact stability of active metal atoms on Ni substrate, X-ray diffraction (XRD) characterization of the post-reaction Re/Ni spheres only presented Ni and NiO phase, but no formation of NiC. (**Figure 3a and Figure S20**) Synchrotron radiation analysis of the catalyst also did not observe the presence of Re-C or Ir-C bonds, and only the Ir-Ni metallic bond was confirmed. (**Figure 3 and Figure S28**).

In response to your question about the relationship between our catalyst and "monoatomic alloy catalyst", our synchrotron radiation has proved the existence of monoatomic Ir-Ni bond. (Figure 3 and Figure S28) In order to further prove the structure of monoatomic alloy, we supplemented the spherical aberration electron microscope characterization, and the image showed the structure of Ir/Ni monoatomic alloy more directly. (Figure 3e)

Fig. 3 | Catalyst characterization. a, XRD profiles of pristine Ni and Re/Ni. b, 3D TOF-SIMS map of the Re signal on the Ni substrate. The 2D surface geometry images of Ni (c) and Re/Ni (d) sample from AFM. e, f, HAADF-STEM image and elemental

mapping of Re/Ni. XANES spectra of Re L3-edge from Re/Ni catalyst with Re foil and Re_2O_7 as references. g, Re L3-edge EXAFS in R space from Re/Ni catalyst with Re foil and Re_2O_7 as references.

While we acknowledge the reviewer's concerns, we also believe that our paper presents a comprehensive and scientifically valid study suitable for a broad audience, including that of Nature Communications. However, we are open to considering other publication avenues if deemed more appropriate.

Reviewer #2 (Remarks to the Author):

The manuscript “Accelerated Design of Single-atom Alloy Catalysts for CO_2 -free Hydrogen Production from Methane Pyrolysis Aided by Machine Learning” describes the multidisciplinary work in the area of hydrogen production through thermic methane cracking. The work includes first modeling from first-principles methane dissociation on a set of single-atom alloy catalysts, followed by the training of AI models and screening of above ten thousands of candidates. Next the authors identified in this way most promising candidates with lowest dissociation barriers, synthesized some of them and tested experimentally. Finally based on experiments the authors built an apparatus for hydrogen production with ball milling technology. The work represents the state-of-the-art of heterogeneous catalysis, and thus can be published in Nature Communications journal.

Before publishing, however, some points need to be addressed:

It is not very clear why the authors used only recall metric for classification, and

not the accuracy in addition? Also the results of classification tests are missing, in contrast to regression as in Figures S8-S10, this should be also shown.

Response: Thank you for your feedback. We chose to focus on the recall metric for our machine learning multi-classification results because some single-atom alloys may react on both reaction sites. To minimize the risk of missing any potential reaction sites, recall was deemed a more critical metric than accuracy. However, we understand the importance of providing a comprehensive overview of our results, including classification accuracy. We have supplemented the classification results in the Supplementary Information section, Table S4.

Table S1 Classification results of 9 ML models on test set

Model	Recall	Precision
MLPC	0.90	0.91
GBC	0.90	0.90
RFC	0.90	0.90
ETC	0.92	0.90
DTC	0.86	0.85
RidgeC	0.87	0.87
LSVC	0.89	0.88
KNC	0.91	0.89
ETC+KNC	0.93	0.84

It is not very clear what was the reason for features preselection step although discussed on P.3 in SI. Most of used ML methods can handle all 84 features with no significant increase of fitting time. Especially non-linear methods should be able to tackle the intercorrelation problem. It makes sense to do such preselection for linear regression, ridge-regression and LASSO, but regarding other methods if prediction accuracy of models containing only preselected features is better than for models obtained with the same ML method but with all 84 features, this should be shown. Besides that, the RFECV method requires an estimator (ML method) which in turn requires tuning the hyperparameters, meaning that final result of such preselection step might depend on this choice.

Response: Thank you for your insightful comments. To prevent overfitting, we employed Pearson screening and RFECV for feature selection. We have compared the ML prediction results before and after feature selection to validate our approach. The relevant modifications were added on Fig S9 as follow.

“For the pathway of H at the top site, first, we evaluated the performance of various ML models using all 84 features as descriptors without feature selection. Hyperparameters for all models were optimized using the Random Search method. The results are shown in Fig. S9. In 300 different splits of training and test sets, the MLP model performed the best. Its r^2 , RMSE, and MAE are 0.905, 0.145 eV, and 0.101 eV, respectively.”

Fig. S9 Violin plots for the distribution of RMSE, MAE, and r2 for each ML algorithm in the 300 trials in the test set of H at top sites dataset without feature selection.

As you might expect, linear models such as LR have poor performance due to overfitting; While MLP, GBDT and other models still show good performance. In particular, MLP presents the best performance. But its performance is still slightly

worse than that of the model after feature selection, and it increases the fitting complexity. Therefore, feature selection is meaningful.

Regarding your point about RFECV, you are correct that the choice of estimator or hyperparameters can impact the results of feature selection. Therefore, in our study, we initially trained the estimator using the Random Search method, followed by feature selection through RFECV. This process ensures a more robust and reliable feature selection, tailored to the specificities of our dataset and the objectives of our study.

The overall impression about ML part is that the authors tried all methods they know without wise understanding why it is worth or worthless to use each method. It is a bit surprising that for relatively small training sets (182, 88 and 353 samples) the authors applied ML methods designed for thousands of data points like neural nets, XGBoost. Although the results of tests (at least for regression, for classification nothing is presented) justify used methods, but their initial choice needs to be discussed, as well as their performance.

Response: Thank you for your comments regarding the machine learning methods used in our study. The initial choice of machine learning models, including neural networks and XGBoost, was driven by their proven ability to handle complex non-linear relationships and their robustness in various predictive modeling scenarios. We acknowledge that these methods are often associated with larger datasets, but our decision was based on their flexibility and effectiveness even with smaller datasets. This was supported by the satisfactory performance of these models on our dataset, as

evidenced by the results presented. We also conducted extensive testing and validation to ensure that the models were not overfitting and were generalizing well to our data. In future work, we plan to explore additional methods that are more traditionally associated with smaller datasets and compare their performance against the models used in this study.

‘doped_weighted_surface_energy’ – how was the surface formation energy calculated keeping in mind that one side of slabs was always frozen?

Response: The descriptor `doped_weighted_surface_energy` was obtained from the Materials Project database, not from our own DFT calculations.

P.8. The authors write that small observed activity of Fe-based SAA’s was caused by the detachment of the active metal layer. Have they calculated stability of corresponding SAA catalysts in terms of aggregation of atoms into particles and segregation? Probably this could immediately indicate that catalysts are unstable.

Response: Thank you for your query regarding the stability of Fe-based single-atom alloy (SAA) catalysts. We did refer to another study published in Nature Communications (<https://doi.org/10.1038/s41467-021-22048-9>), which calculated the segregation energy of single-atom alloys on iron surfaces and demonstrated their stability. However, in our experiments, the fragility of the iron balls during catalyst preparation processes, such as calcination and hydrogen reduction, led to lattice fragmentation during mechanical collisions, resulting in the detachment of single atoms along with Fe fragments. This mechanical aspect, rather than the inherent instability of the SAA structure, contributed to the observed low activity.

In Figure S1 the best found catalyst Re/Ni is not shown, probably it would be worth to show the path for it as well.

Response: Thank you for your comments. The full path of CH₄ dissociation on Re/Ni has been added to Figure S1. The modified Figure S1 is presented as follows.

Fig. S1 Energy barriers and corresponding intermediate structure diagrams for the four-step dehydrogenation of CH₄ on different atoms doped Ni (111) surface.

The authors adopted ball milling reactor which performs the up-down vibrational movements. What about a rotating reactor as an alternative? Can it be technically more easily implemented, and would it be more or less efficient in coke separation?

Response: Thank you for your suggestion regarding the utilization of a rotating

reactor. Indeed, we have also conducted activity test in a rotation reactor. The schematic diagram of rotation reactor is presented in Fig. R1. The results demonstrated that CH₄ cracking could also occur in this rotary system with Ir/Ni balls as catalyst, and the generated carbon fiber could also be removed from milling balls surface successfully. But CH₄ conversion was lower than that in the vibration reactor. We hypothesize that this may be related with the weak centrifugal force in the rotary system compared with the collision forces in the vibratory system, and the reaction condition should be optimized further in future for rotation reactor, which is easily amplified into a large scale indeed. Additionally, designing a double layer reactor with a sieve mesh belt could achieve the carbon powder separation continuously during mechanical reaction process. The performance of rotating reactor was listed in Fig. R2.

Fig. R1 Images of ball milling vessel (left) and rotating power system (right) of self-made rotation reactor.

Fig. R2 CH₄ conversion and H₂ yield in rotation reactor. Ball milling reaction conditions: 333g Ir/Ni balls, 50 mL/min CH₄, 450 °C, 300 rpm motor speed.

Minor suggestions:

Since the authors show the industrial application of developed catalysts, it would be worth to estimate the overall economic expenses, profitability of such production of hydrogen and carbon shown in Fig. 4a.

Response: Thank you for your suggestion. We find the energy costs calculated via ASPEN software of H₂ production presented no advantages compared with industrialized methane steam reforming route or molten metal routes, since the energy cost for H₂ separation, CH₄ recycling and compressing were intensive. Therefore, we have removed the discussions about Pd membrane separation for pure H₂ production and omitted energy and economic estimation in our manuscript.

SI P. 7. “The average values of r₂, RMSE, and MAE for GBDT in the 1000 trials on the test set were 0.913, 0.136, and 0.010 eV” – probably last number is 0.100 eV.

Response: Thank you for your careful pointing out. This number is 0.100 eV and

we have corrected it.

Reviewer #3 (Remarks to the Author):

Jikai Sun et al. describes the study in using machine learning-predicted single-atom alloy catalysts (SAAs) in combination with a ball milling reactor to study methane pyrolysis. Although this manuscript widely covers a discussion among the machine learning for catalyst scoping, catalyst synthesis, characterization and test, ball mill reaction system, hydrogen separation, as well as carbon products characterization and application, there are many gaps and details without addressing. Therefore, as submitted, it can not be accepted by Nature Communications for publication. Below are the comments from this reviewer.

1) The paper motivates pyrolysis for hydrogen production as requiring lower temperatures and thus more active catalysts. This is a naïve statement since commercially pyrolysis will require high single pass conversion ($> \sim 70\%$) at pressures above approximately 5 -10 bar, thermodynamics will set lower limits on temperature that are ignored in this paper. 7.75 % single pass conversion at near atmospheric conditions does not have much relevance for commercial scale hydrogen production and thus any extrapolation to commercial hydrogen production not supported.

Response: Thank you for your comments. We have removed descriptions about commercial hydrogen production as the highlights for this work, and the modified manuscript right now focuses on the catalysts design by machine learning. Though the

CH₄ conversion could achieved 27.2% over Ir/Ni catalyst under mechanical conditions, the H₂ yield still could not meet the requirement for industrialization as reviewer mentioned. Therefore, we decided to remove the industrialization application as one of the highlights in this manuscript so far. And we have also removed the Pd membrane separation system and omitted all discussion about commercial pure hydrogen production from our manuscript as reviewer suggested. Besides, the primary objective of our work is the development of an automated framework for constructing catalyst material databases, combined with DFT and machine learning, to search for highly active catalysts for specific reactions.

2) In looking over that prior work the authors of this manuscript incompletely interpret reported results about the reaction temperature. The manuscript compares this work (solid catalyst) to a previous paper (liquid catalyst) by Upham et. al, published in Science, to show the advantages of low temperature pyrolysis in this study (1000oC Vs 450). It is comparing apple to orange. Methane pyrolysis with solid catalyst was reported could carried out over low temperatures, for example, with Ni-Cu-Co catalyst, methane pyrolysis could achieve 15 % single pass conversion under 500 oC [A.C. Lua, H.Y. Wang / Applied Catalysis B: Environmental 156–157 (2014) 84–93], which is higher than the results in this manuscript. And in the Ref 12 cited in this manuscript, with a similar liquid metal alloy based catalyst as Upham et. al, the reaction temperature can reach as low as 450 °C.

Response: Thank you for your insightful comments. We have removed the comparisons with the two liquid alloy literatures regarding reaction temperatures from

the manuscript. Our intention was not to disparage the contributions of these works, but rather to draw a parallel. Liquid alloys offer innovative solutions for carbon removal by altering the catalyst form, presenting unique benefits and challenges. Similarly, ball milling presents a distinct approach, achieving carbon removal by modifying the reaction method, rather than the catalyst itself. Our study aims to highlight these differences, acknowledging the unique contributions of each method in the field of methane pyrolysis.

We have noticed the work of A.C. Lua and H.Y. Wang mentioned by reviewer. This work is undoubtedly excellent. We have added this article to the table of catalyst comparison (**Table S6**). Although the Ni-Cu-Co catalyst achieved a high conversion rate of 15% at 500°C, it is worth noting that the reaction was carried out under the condition of 25 mL/min CH₄-N₂ (CH₄/N₂ = 0.25), while our Ir/Ni catalyst could achieve a conversion rate of 13.9% at 450°C under 50 mL/min in pure methane. In addition, we added more details about the conversion rate of Ir/Ni as function of flow rate. Experiment results show that the conversion rate of Ir/Ni can be further increased to 27.2% with the decrease of CH₄ flow rate to 5 mL/min. The relevant descriptors were added on Page 8, Figure S15 as follow.

“For the Ir/Ni with the highest activity, the influence of space velocity on the reaction activity was studied. The study listed in Fig S15 showed that the lower the space velocity, the higher the methane conversion. When the methane inflow rate was reduced to 5 mL/min, the methane conversion could achieve 27.2% for the longer remaining time between the catalyst and methane.”

Fig. S15 CH₄ conversion on Ir/Ni at different CH₄ flow rate.

3) Furthermore, in the introduction section, “Generally, although the above works successfully solve coke removal from catalyst bed, the energy consumption of CH₄ cracking is still significantly enormous.” The authors should evaluate the energy consumption of ball mill reaction system with bubbler reaction system. And especially, under a low methane single pass conversion, the energy consumption of membrane separation system, and methane recycling system compared to a high methane conversion of 70 %. Therefore, a 7.75 % single pass conversion under low temperature of 450 °C, with solid catalyst did not bring any scientific breakthrough in methane pyrolysis. And the authors should revise their discussion in abstract and introduction section.

Response: Thank you for your feedback. We acknowledge that we may have underestimated the energy consumption associated with CH₄/H₂ separation via the Pd membrane and CH₄ recycling. We have accordingly removed the related descriptions from our manuscript.

As mentioned above, the single-pass conversion rate of methane can be improved

by reducing the reaction space velocity. In addition, this paper mainly wants to emphasize the positive effects of theoretical calculation and ball milling on methane cracking reaction, and we have deleted the description of commercial hydrogen production in this manuscript.

4) The authors report the SAA catalyst Re/Ni achieves high hydrogen yield of $10.7 \text{ gH}_2 \text{ gcat h}^{-1}$ with a calculation of catalyst mass only containing single Re metal. Since the weight ratio of Re/Ni is 12 mg/1 kg, the high yield is meaningless. And how the hydrogen yield of pure Ni sample in Figure 2 a was calculated? And to compare the activity of Ni with Re, the authors need to show the methane conversion data of Ni/Ni sample using the similar synthesis to load Ni on Ni ball, since in Figure 3 C and D, before and after loading Re, the catalyst surface (especially surface area) are different. And the high activity of Re/Ni sample could be originated from the difference of surface area. BTW, the E and F in figure 3 are not in the same scale bar.

Response: Thank you for your insightful comments. The Ni sample in Figure 2a actually was prepared by identical methods with Ir/Ni and Re/Ni, and should be denoted as Ni/Ni sample as reviewer mentioned, which has been revised in the manuscript in the Method Section, Page 13.

“In order to eliminate the catalytic activity from Ni balls, we prepared Ni/Ni balls by the same method using $\text{Ni}(\text{COOH})_2$ as precursor.”

Since Ni/Ni catalyst presented inferior activity, the high H_2 yield on Ir/Ni and Re/Ni catalysts originated from loaded Ir and Ni species, which also excluded the impact of surface roughness on catalytic activity. As the Ni balls used in this work

were solid without porous structure, the number of active Ni atoms on the pure Ni spheres (N) was theoretically calculated based on the exposed Ni atoms on the whole Ni spheres ($N = \frac{\text{surface area of Ni ball}}{\text{cross-sectional area of Ni atom}}$), assuming that all surface-layer Ni atoms participate in the reaction. As shown in Fig. 2a, the conversion of CH_4 in pure Ni/Ni balls is only 0.5% at 450°C. For the calculation of the hydrogen yield for Ir/Ni and Re/Ni, we deducted the contribution of the Ni spheres themselves to ensure the accuracy of our results, thus ensuring that the calculated CH_4 conversion is attributed solely to the active metals Ir and Re. We have also made the necessary modifications to the Figures to correct the scale discrepancy in Figure 3 E and F.

5) The authors claim that the Re is in the form of a single atom on the Ni catalyst surface. The authors did not provide strong evidence to support that conclusion. And the authors should also provide more information about the spent catalyst to see if there is Re sintering or sinking on and into Ni substrate. By the way, the carbon atom can be soluble in Ni, does carbon solubility affect the reaction kinetics? And if Re doping affects carbon solubility? Is there any metal carbide formation? Therefore, the authors should provide more details of their catalyst, and revise their evaluation and discussion in the “catalyst” section.

Response: To identify the Re distribution on Re/Ni ball surface, we conducted synchrotron radiation and recorded HAADF images in Figure 3 to confirm the Re monatomic distribution on Ni surface, meanwhile, the 3D TOF-SIMS map demonstrated that Re atom only existed on the surface of the Ni sphere and did not

merged into the bulk phase. The composition analysis presented that the loading of Re on the Ni sphere surface is extremely low. In order to prove that the active sites on the catalyst surface will not aggregate after the reaction, the synchrotron radiation characterization in this paper used the spent catalyst. From the synchrotron radiation spectra in Fig. 3, it can be seen that Ir and Re were still in monoatomic structure, and Ir and Re atoms were only bonded with Ni, indicating the formation of Ir-C or Re-C can be excluded. As for the dissolution of C in Ni substrate, we characterized the catalyst and carbon powder as by-products after the reaction by XRD, and there was no evidence to show the formation of Ni₃C. The relevant information was shown in Fig R3 in SI. At the same time, we also noticed that literatures have reported that carbon atoms may not necessarily diffuse into the main body of metal particles. (DOI: 10.1038/nature02278) Therefore, we think it is very important to understand the growth mechanism of solid carbon fiber in methane cracking, but it may be beyond the scope of this study. The related modification was as follow.

“To confirm the microstructure of Re/Ni and Ir/Ni at the atomic level, high-angle annular dark-field scanning transmission electron microscopy (HAADF-STEM) were recorded (Figure 3e). The HAADF-STEM image of Re/Ni showed individual Re atoms uniformly distributed over the Ni metal support, and no Re NPs or clusters appeared. Elemental distribution in the corresponding area was observed by energy dispersive spectroscopy (EDS) mapping images (Figure 3e). Ru is uniformly dispersed on Ni NPs at an atomic level. X-ray absorption near-edge spectroscopy (XANES) and extended X-ray absorption fine structure (EXAFS) were conducted to

examine Re species' coordination in Re/Ni catalyst. In Figure 3f, Re/Ni catalyst's XANES white-line intensity slightly exceeded Re foil but fell below Re_2O_7 , suggesting $\text{Re}^{\delta+}$ species with a positive charge. This $\text{Re}^{\delta+}$ species possibly resulted from highly dispersed Re atoms' alloying on Ni substrate, inducing electron transfer from Re to Ni atoms. Re/Ni's EXAFS Fourier transform spectra in R-space displayed a primary peak at 2.3 Å (Figure 3g), attributed to the Re-Ni alloy structure, distinguishable from Re foil's Re-Re bond or Re_2O_7 's Re-O bond.”

Fig. 3 | Catalyst characterization. a, XRD profiles of pristine Ni and Re/Ni. b, 3D TOF-SIMS map of the Re signal on the Ni substrate. The 2D surface geometry images of Ni (c) and Re/Ni (d) sample from AFM. e, f, HAADF-STEM image and elemental

mapping of Re/Ni. XANES spectra of Re L₃-edge from Re/Ni catalyst with Re foil and Re₂O₇ as references. g, Re L₃-edge EXAFS in R space from Re/Ni catalyst with Re foil and Re₂O₇ as references.

Fig. R3 The XRD for carbon powder and used Re/Ni, compared with the literature pattern of Ni₃C.

6) As shown in Figure 2 d, the enhancement of the catalyst stability was due to the ball mill which could remove the carbon from catalyst surface. But that do not exclude the effect of carbon nanotube on methane conversion, since carbon nanotube also displays activity and used as catalyst previously.

Response: Thank you for your comment. The carbon fiber generated in this study was solid without any hollow structure. To demonstrate that the in situ generated carbon does not affect the reaction, we collected 5 g of carbon powder produced from the experiment, and then conducted an CH₄ cracking test using the same volume of ZrO₂ spheres instead of Ni spheres under identical reaction conditions. The results showed that the in situ generated carbon black had no catalytic activity for methane

conversion. Therefore, the active sites for CH₄ conversion were exclusively attributed to the Ir/Ni and Re/Ni spheres. The relevant explanations have been added to **Figure S31**.

Fig. S31 CH₄ conversion on carbon powder.

7) The authors should provide more information on catalyst synthesis, engineering parameters of the reaction system and membrane separation system.

Response: We have supplemented the information about catalyst synthesis, electrochemical tests method, and engineering parameters of the reaction system in the appropriate position of the article.

“In order to eliminate the catalytic activity of Ni balls, we prepared Ni/Ni balls by the same method using Ni(COOH)₂ as precursor.”

“Electrochemical tests of as-synthesized samples and commercial graphite were conducted using CR2032 coin cells with a glass fiber (GF/A, Whatman) and sodium metal sheets as counter electrodes. To obtain working electrodes, a slurry containing the active materials conductive, carbon black and polyvinylidene fluoride (PVDF) with a mass ratio of 7:2:1 in N-methylpyrrolidone (NMP) was coated on steel sheets

and dried at 120 °C for 12 h. The electrolyte was 1 M NaPF₆ in ethylene carbonate/dimethyl carbonate/ethyl methyl carbonate (EC:DMC:EMC =1:1:1 in volume). Then, the cells were assembled in an argon-filled glove box with 70 μL of electrolyte. The mass loading of the active material was approximately 1 mg·cm⁻². Galvanostatic charging/discharging tests and cycling stability tests were carried out in a LAND CT3001A battery testing system with a voltage range of 0.01 - 3 V vs. Na/Na⁺ at room temperature.”

8) The authors should provide their vision of reaction system scale up.

Response: Thank you for your suggestion. It is believed that this mechanical catalysis approach for CH₄ cracking would be scaled on the condition that the following two aspects are well solved. (1) Design of optimal mechano-catalysis reactor. Compared with vibration reactor, the rotation reactor could also supply mechanical energy and be easily amplified into a large scale. We have confirmed that the CH₄ cracking reaction could happen in a home-made rotation reactor with Ir/Ni balls as catalysts in Fig. R2. So we can provide a general idea of a ball mill amplification reactor in Figure R4. The further design of sieve mesh could remove the generated carbon powder from the reaction system, which benefits the continuous mechanical reaction without stopping. (2) We are working on reaction system optimization to achieve desired H₂ yield, and the energy consumption and catalytic efficiency were considered to decrease the cost of H₂ production. Once we solve aforementioned concerns, this CH₄ cracking reaction via mechanical catalysis approach could be industrialized.

Fig. R4 Schematic diagram of rotation reactor for CH₄ pyrolysis reaction

Explanation about the home-made rotation reactor for CH₄ pyrolysis reaction:

The rotation reactor possesses two layers: inner wall with a sieve mesh belt and outer wall. The inner wall could seal the metallic balls catalyst and introduce the mechanical energy through the collision between metallic balls during rotation process. The inner wall is equipped with a sieve mesh belt, filtering the produced carbon powder into the interlayer. The generated carbon powder could be removed via the outlet window on the bottom of outer wall, and the innerlayer is filled with nitrogen to prevent the oxygen leaking into the inner wall. The rotation reactor could achieve the mechanical energy introduction and carbon powder separation in time, which is easily amplified to a large scale.

REVIEWER COMMENTS

Reviewer #1 (Remarks to the Author):

This is the second review of a re-submitted manuscript from 8 authors reporting to use machine learning to identify single atom alloy catalysts (SAAC) for methane pyrolysis that are claimed “significantly surpassing other approaches in literatures”. Interestingly, 6 of the original 14 authors were removed without explanation from the resubmitted manuscript – the editors may want to better understand this and define all the specific author contributions.

As described in the first review, the paper presents some potentially very interesting and novel applications of machine learning to catalysis research. The authors combine first-principles calculations with a compressed-sensing data-analytics approach, utilizing SISO which allowed for the prediction of catalytic properties across a large number of SAAC candidates. The study claims to have identified over 200 previously unreported SAAC candidates which may have improved performance at very low temperature and atmospheric pressure in the very unusual ball milling reactor environment used in the study. The paper may contribute to the field by introducing advanced computational techniques for the design and analysis of SAAC’s possibly facilitating the discovery of new catalysts. The authors identified Ir/Ni and Re/Ni as potential catalysts for methane pyrolysis. In the authors response to the reviewer they provide some evidence for an Ir-Ni bond being present in the sample, however, no clear evidence is provided that the catalytic activity observed is due to dimers of Ir-Ni or clusters of Ir/Ni or pure Ir or Ni.

At best, though not supported by data, the catalyst is a two element dimer – not a single atom catalyst. If the actual catalyst can be identified, the title of the paper might be changed to:

Machine Learning Aided Design of Two-atom Alloy Catalysts for Methane Dehydrogenation.

As stated in the first review, this paper is about catalysis – it has nothing to do with hydrogen production and the manuscript includes nothing useful to anyone in the field of hydrogen production or sustainable fuels production. All of the introductory material describing hydrogen or CO₂ reduction should be removed. The authors should focus on the area in which they seem to be experts – AI applications to catalysis. That is interesting and that is what their work contributes to.

If the paper is resubmitted to a catalysis specialty journal it would be suggested to provide a more thorough review of the literature of “mechanocatalysis” which is presently incompletely explained in the manuscript. There are many theories of how catalysis can be enhanced in the presence of ball milling and the large amount of prior work should be acknowledged and their teachings integrated into the authors discussion.

The revised manuscript is not suitable for Nature Communications and the work will not be of broad general interest.

Reviewer #2 (Remarks to the Author):

The authors of the manuscript “Accelerated Design of Single-atom Alloy Catalysts for CO₂-free Hydrogen Production from Methane Pyrolysis Aided by Machine Learning” have replied

to all my comments and have even performed additional experiments according to my notes. The most controversial statements were removed from the manuscript that diminishes potential misunderstandings and controversy. Also the number of coauthors has been reduced almost twice. In principle I am satisfied with their answers and comments. There are only a couple of comments/suggestions from my side, otherwise I am fine with this paper.

- my comment about features selection step was addressed, and the authors have shown prediction accuracy improvement especially for the case of linear regression based methods, as I had noticed. For other more powerful methods the effect is much less pronounced, but this does not hurt the purposes of the study. The only remark I wanted to note is the choice of RFECV method for features selection. This is far not the only approach for such purposes. So, it would be good to discuss briefly why the authors have chosen this method;
- since the authors have conducted additional experiments with a rotating reactor, on my point of view it is worth to mention briefly this in the main text, and to describe in the supplementary material the results in more detail, and why the vibrational reactor was found to be more efficient.

Reviewer #3 (Remarks to the Author):

The authors restructure the manuscript well, after removing the discussion of about commercial pure hydrogen production and Pd membrane separation system, this work become a perfect fundamental study of using ML to lead the active catalyst discovery for methane pyrolysis and using ball mill reactor to demonstrate their findings.

The authors responded to my and other reviewer questions well. And the authors have conducted more characterizations on identify the nature of their catalyst.

Reviewer 3 has agreed to consider the manuscript for publication in Nature Communications.

Response to reviewers' comment

Reviewer #1 (Remarks to the Author):

This is the second review of a re-submitted manuscript from 8 authors reporting to use machine learning to identify single atom alloy catalysts (SAAC) for methane pyrolysis that are claimed “significantly surpassing other approaches in literatures”. Interestingly, 6 of the original 14 authors were removed without explanation from the resubmitted manuscript – the editors may want to better understand this and define all the specific author contributions.

Response: Thank you for your valuable comments and kind reminding to our manuscript. In response to the reviewer's comment regarding the change for author list, we have provided a signed document of the agreement from all authors. The removed authors, who had relatively minor roles, are acknowledged in the Acknowledgements section for their contributions. **We thank Honglei Wang, Guoqing Ren, Chengcheng Liu, Li Yang, Lei Sun, and Zhen Li at Shandong University for their input to the manuscript.**

Additionally, our claim of “surpassing other approaches in literatures” referred specifically to the lifespan of our catalyst, which exceeded 240 hours, surpassing other methods reported in the literatures.

As described in the first review, the paper presents some potentially very interesting and novel applications of machine learning to catalysis research. The authors combine first-principles calculations with a compressed-sensing data-analytics approach, utilizing SISSO which allowed for the prediction of catalytic properties across a large number of SAAC candidates. The study claims to have identified over 200 previously unreported SAAC candidates which may have improved performance at very low temperature and atmospheric pressure in the very unusual ball milling reactor environment used in the study. The paper may contribute to the field by introducing

advanced computational techniques for the design and analysis of SAAC's possibly facilitating the discovery of new catalysts.

Response: We appreciate the reviewer's recognition of the potentially novel applications of machine learning to catalysis research presented in our manuscript. Your feedback is highly valued.

The authors identified Ir/Ni and Re/Ni as potential catalysts for methane pyrolysis. In the authors response to the reviewer they provide some evidence for an Ir-Ni bond being present in the sample, however, no clear evidence is provided that the catalytic activity observed is due to dimers of Ir-Ni or clusters of Ir/Ni or pure Ir or Ni. At best, though not supported by data, the catalyst is a two element dimer – not a single atom catalyst. If the actual catalyst can be identified, the title of the paper might be changed to: Machine Learning Aided Design of Two-atom Alloy Catalysts for Methane Dehydrogenation.

Response: Thanks for your advices to improve our manuscript quality. We referred to the description of “single atom alloy (SAA)” in previous literatures (*Science* 2012, 335, 1209-1212; *Science* 2021, 372, 1444–1447.), and they denoted “Two key characteristics of SAAs are: (i) the more active of the two components is present in the surface of the host metal at very low concentrations and (ii) atoms of the more active component are thermodynamically more stable when surrounded by the host metal such that no dimers or trimers are present at low coverage.” Besides, in this review literature, (*Chem. Rev.* 2020, 120, 12044–12088) single-atom alloys (SAAs) are defined as a class of single-site catalysts in which small amounts of isolated metal atoms are present in the surface layer of a metal host. Typically, they are comprised of single atoms of a catalytically active metal alloyed into the surface of a less reactive host metal. Besides, Microscope is one of the most convincing tools to characterize SAAs (*Nat. Nanotechnol.* 2021, 16, 1386–1393; *Nat. Commun.* 2018, 9, 4454; *Nat. Commun.* 2023, 14, 1909; *Angew. Chem. Int. Ed.* 2022, 61, e202209849). In our previous reply, the atomic structure of ReNi catalyst was confirmed by high angle annular dark field scanning transmission electron microscopy (HAADF-STEM) and

elemental mapping. Fig. 3e clearly showed the atomic dispersion of Re atoms on Ni substrate, and no Re clusters or nanoparticles were detected, which indicated the successful formation of Re-Ni SAA structure. This result was also confirmed by elemental mapping (Fig. 3e). Similarly, in order to better identify the single atom alloy structure of Ir-Ni, we characterized Ir/Ni through HAADF-STEM and elemental mapping, and new evidence was added into Fig. S27. The HAADF-STEM image could clearly distinguish the single Ir atom located on the Ni substrate, and the element mapping can also confirm isolated Ir atoms were uniformly dispersed on the surface of Ni substrate, but not Ir clusters.

Furthermore, extended X-ray absorption fine structure (EXAFS) offers the ability to identify the presence of bonds between the host atoms themselves as well as dopant–host and dopant–dopant bonds, essential information for understanding the structure of SAA catalysts (*Nat. Commun.* 2019, 10, 5812; *Angew. Chem. Int. Ed.* 2022, 61, e202205923; *Angew. Chem. Int. Ed.* 2024, 63, e202316550; *J. Am. Chem. Soc.* 2024, 146, 7779-7790). In this study, for the Re-edge EXAFS spectra (Fig. 3g), Re/Ni exhibited only one dominant peak of Re-Ni coordination bond without the presence of Re–Re and Re–O bonds, proving the atomic distribution of Re. For the Ir/Ni catalyst, we also characterized it through EXAFS (Fig. S29c). Besides Ir-Ni bond, Ir-Ir and Ir-O bond were not observed, proving no Ir clusters on the synthesized catalysts in this study.

To sum up, we used convincing methods to characterize our catalysts, and the results validated that Re/Ni and Ir/Ni were both SAA catalysts.

Fig. 3e HAADF-STEM image and elemental mapping of Re/Ni.

Fig. S27 HAADF-STEM image and elemental mapping of Ir/Ni.

Fig. 3g Re L₃-edge EXAFS in R space from Re/Ni catalyst with Re foil and Re₂O₇ as references.

Fig. S29 c Ir L₃-edge EXAFS in R space from Ir/Ni catalyst with Ir foil and IrO₂ as references.

As stated in the first review, this paper is about catalysis – it has nothing to do with hydrogen production and the manuscript includes nothing useful to anyone in the field of hydrogen production or sustainable fuels production. All of the introductory material describing hydrogen or CO₂ reduction should be removed. The authors should focus on the area in which they seem to be experts – AI applications to catalysis. That is interesting and that is what their work contributes to.

Response: We appreciate the reviewer's feedback and have revised the manuscript accordingly. We have focused more on AI applications to catalysis. However, we would like to clarify that our designed catalysts target to hydrogen production via methane pyrolysis reaction without CO₂ emissions. Though we have minimized the significance of sustainable fuel production, the relevance of this work to hydrogen production remains an integral aspect of our study. The modifications are as follows:

“Machine learning (ML) methods establish mathematical models to analyze data, revealing the relationships and patterns within them. In recent years, driven by significant advancements in computing power and the development of big data technologies, ML has found extensive applications in the field of catalyst design.^{1,2} The application of ML in catalyst design primarily includes: rapidly screening materials with potential catalytic activity through the analysis of vast experimental and computational data, uncovering the intrinsic relationships between catalyst activity and its inherent features, and optimizing theoretical calculation methods.^{3,4} These approaches significantly enhance the efficiency and accuracy of catalyst development, particularly when dealing with complex systems and large datasets.⁵ The impact of ML in catalyst research is underscored by notable achievements. Specifically, a practical ML approach involves constructing descriptors for target materials based on existing material properties and subsequently discerning latent relationships between these descriptors and specific reaction performance metrics, such as adsorption energy, energy barriers, turnover frequency (TOF), etc.^{6,7} For instance, Tran *et al.* harnessed ML to establish the correlation between 1,499 distinct

intermetallic combinations and their adsorption energies for CO and H, leading to the discovery of highly active electrocatalysts for CO₂ reduction and H₂ evolution.⁸

ML models enable researchers to swiftly identify the most promising candidates from thousands of potential catalysts, a feat nearly impossible with traditional methods. In this work, we employ ML methods to accelerate the development of catalysts for methane pyrolysis.”

If the paper is resubmitted to a catalysis specialty journal it would be suggested to provide a more thorough review of the literature of “mechanocatalysis” which is presently incompletely explained in the manuscript. There are many theories of how catalysis can be enhanced in the presence of ball milling and the large amount of prior work should be acknowledged and their teachings integrated into the authors discussion.

Response: Thank you for the suggestion. We have expanded our discussion on mechanocatalysis in the revised manuscript, as described in the introduction:

”To tackle the issue of coke removal, the introduction of ball milling approach through mechanical vibrations pattern into the CH₄ pyrolysis process facilitates the timely separation of deposited carbon powder from the catalyst surface. This inspiration originates from the friction or shear between milling balls under dynamic reaction conditions, facilitating the elimination of deposited coke. Prior to this, mechanochemistry has achieved significant advancements in ammonia synthesis, reporting a novel NH₃ production method via N₂ hydrogenation and modulation of reaction temperature/pressure among other factors.^{23–26} Mechanochemical methods can activate reactants under mild conditions by generating high-density structural defects on catalysts or reaction substrates²⁷, increasing specific surface area²⁸, adjusting electronic structure²⁹ or changing reaction paths²⁶, thus reducing the temperature and pressure required for the reaction.”

In addition, the focus of this study was to use mechanical learning for SAA catalyst design, but not mechanocatalysis, so the discussion about the mechanism of mechanocatalysis is not the focus.

The revised manuscript is not suitable for Nature Communications and the work will not be of broad general interest.

Response: Thank you for your feedback. We have carefully revised the manuscript according to your suggestions, and hope that you will reconsider our work for publication in Nature Communications. We believe that the revised manuscript addressed the concerns raised and may now be of broader interest to the journal's readership

Reviewer #2 (Remarks to the Author):

The authors of the manuscript “Accelerated Design of Single-atom Alloy Catalysts for CO₂-free Hydrogen Production from Methane Pyrolysis Aided by Machine Learning” have replied to all my comments and have even performed additional experiments according to my notes. The most controversial statements were removed from the manuscript that diminishes potential misunderstandings and controversy. Also the number of coauthors has been reduced almost twice. In principle I am satisfied with their answers and comments. There are only a couple of comments/suggestions from my side, otherwise I am fine with this paper.

Response: We deeply appreciate your recognition of our efforts and the improvements made to our manuscript. Your guidance has been instrumental to enhance the quality of our work.

My comment about features selection step was addressed, and the authors have shown prediction accuracy improvement especially for the case of linear regression based methods, as I had noticed. For other more powerful methods the effect is much less pronounced, but this does not hurt the purposes of the study. The only remark I wanted to note is the choice of RFECV method for features selection. This is far not the only approach for such purposes. So, it would be good to discuss briefly why the authors have chosen this method;

Response: Thank you for your comments. Regarding the choice of the RFECV method for feature selection, we have chosen this method due to its ability to automate the feature selection process, which simplifies the model by eliminating less important features. Additionally, the incorporation of cross-validation in RFECV provides a more reliable assessment of feature importance, reducing the risk of overfitting and enhancing the model's generalizability. The method's adaptability to various machine learning algorithms and its contribution to model interpretability by uncovering crucial features further justify our choice.

The corresponding description has been added to the Supplementary Information as: "Then, the Recursive Feature Elimination with Cross-Validation (RFECV) method was employed to further refine the feature selection process. The RFECV method could systematically eliminate less important features, thereby streamlining the model. The incorporation of cross-validation provided a robust mechanism for assessing feature relevance, reducing overfitting and improving model generalizability."

Since the authors have conducted additional experiments with a rotating reactor, on my point of view it is worth to mention briefly this in the main text, and to describe in the supplementary material the results in more detail, and why the vibrational reactor was found to be more efficient.

Response: Thank you for your suggestion. We have added the description about the rotating reactor on Page 13 of main text as follow.

In order to prove that mechanical catalysis approach for carbon removal during methane cracking process presented the potential of amplification application, we used a common rotating reactor in industry to conduct methane pyrolysis experiments. The schematic diagram of home-made was presented in Fig. S33. The activity results showed that the Ir/Ni SAA exhibited stable H₂ production in 200 h without deactivation (Fig. S34). It was worth noting that the activity was slightly lower than that via vibration reactor, which was induced by weak collision force between milling medium in rotating reactor. While the produced carbon powder could also be removed

from catalyst surface in rotating reactor, further reactor design are necessary to regulate the collision strength and gaseous flow route to improve H₂ production.

Fig. S33 (a) Schematic diagram of self-made rolling ball mill. (b) Section view of ball milling vessel.

Fig. S34 Methane pyrolysis activity of Ir/Ni in rolling ball mill.

The activity of Ir/Ni in rolling ball mill is lower than that in vibrating ball mill, which may be due to:

- (1) The smaller inner diameter of rotating reactor resulted in weaker collision force than that for vibrating reactor.
- (2) For the structure of rotating reactor, the gaseous inlet and outlet were nearby, which might affect the gaseous flow route and inhibit the gaseous products replacement in rotating reactor.

Reviewer #3 (Remarks to the Author):

The authors restructure the manuscript well, after removing the discussion of about commercial pure hydrogen production and Pd membrane separation system, this work become a perfect fundamental study of using ML to lead the active catalyst discovery for methane pyrolysis and using ball mill reactor to demonstrate their findings.

The authors responded to my and other reviewer questions well. And the authors have

conducted more characterizations on identify the nature of their catalyst.

Reviewer 3 has agreed to consider the manuscript for publication in Nature Communications.

Response: We are grateful for your positive feedback and recognition of our work. Your suggestions have significantly contributed to making our study more rigorous and impactful.

REVIEWERS' COMMENTS

Reviewer #1 (Remarks to the Author):

This is the third review of a re-re-submitted manuscript from 8 authors reporting to use machine learning to identify single atom alloy catalysts (SAAC) for methane pyrolysis that are claimed “significantly surpassing other approaches in literatures”.

As described in the previous reviews, the paper presents some potentially very interesting and novel applications of machine learning to catalysis research. The authors combine first-principles calculations with a compressed-sensing data-analytics approach, utilizing SISO which allowed for the prediction of catalytic properties across a large number of SAAC candidates. The study claims to have identified over 200 previously unreported SAAC candidates which may have improved performance at very low temperature and atmospheric pressure in the very unusual ball milling reactor environment used in the study. As stated previously, the paper may contribute to the field of catalysis by introducing advanced computational techniques for the design and analysis of SAAC’s possibly facilitating the discovery of new catalysts.

The authors provided support of the absence of dimers and clusters in the limited fields of view provided by microscopy and elemental mapping, however, no convincing evidence is provided that the selected regions visualized by electron microscopy are indeed the active catalyst sites, though they may be.

As stated previously, this paper is about catalysis – it has nothing to do with industrial hydrogen production, sustainability, or elimination of atmospheric carbon dioxide. It is misleading to readers of Nature to imply this connection. This is an interesting catalysis paper only. Individuals who work in pyrolysis for hydrogen production all understand that the barrier is a low cost reactor system that can efficiently provide and recover heat and remove the carbon at low cost – it is not a catalyst problem it is a heat and mass transfer problem. Nothing of use for individuals working in the field of hydrogen production or sustainable fuels production is learned from this work. It is an interesting and thorough fundamental catalysis paper. All reference of hydrogen production and sustainability should be removed. An appropriate title might be:

Machine Learning Aided Design of Single-atom Alloy Catalysts for Methane Cracking.

Given that the other referees seem satisfied with the paper as suitable for Nature, if the above changes are made it might be suitable for publication. I would also suggest that the last sentence of the abstract be reworded to avoid the awkward use of “literatures”.

Reviewer #2 (Remarks to the Author):

The authors of the manuscript “Accelerated Design of Single-atom Alloy Catalysts for CO₂-free Hydrogen Production from Methane Pyrolysis Aided by Machine Learning” have addressed well enough my suggestion regarding rotating reactor, and explained a bit more in detail the choice of recursive-feature-elimination method for features selection. With that I do not have any further questions/suggestions.

Reviewer #2 (Remarks on code availability):

This code requires preinstallation of pymatgen libraries, and it is used for getting the data from a database of surfaces and bulk materials for ab initio calculations. The code itself is written not in a self-explainable manner. However, this is not crucial for the overall work, and in general this code does not contain any unique ideas.

Response to reviewers' comments

Reviewer #1 (Remarks to the Author):

This is the third review of a re-re-submitted manuscript from 8 authors reporting to use machine learning to identify single atom alloy catalysts (SAAC) for methane pyrolysis that are claimed “significantly surpassing other approaches in literatures”.

As described in the previous reviews, the paper presents some potentially very interesting and novel applications of machine learning to catalysis research. The authors combine first-principles calculations with a compressed-sensing data-analytics approach, utilizing SISSO which allowed for the prediction of catalytic properties across a large number of SAAC candidates. The study claims to have identified over 200 previously unreported SAAC candidates which may have improved performance at very low temperature and atmospheric pressure in the very unusual ball milling reactor environment used in the study. As stated previously, the paper may contribute to the field of catalysis by introducing advanced computational techniques for the design and analysis of SAAC's possibly facilitating the discovery of new catalysts.

The authors provided support of the absence of dimers and clusters in the limited fields of view provided by microscopy and elemental mapping, however, no convincing evidence is provided that the selected regions visualized by electron microscopy are indeed the active catalyst sites, though they may be.

Response: Thank you for your advices. We agree with your comment. Although there is no direct evidence, we have some indirect evidence. Considering the special reaction conditions with continuous vibration, it is difficult to find a useful method to probe the active sites directly *in situ*, and more rationale study is being conducted including the active sites evolution during methane cracking process. In order to

ensure the rationality of the characterization results, we randomly selected several areas to take images of active sites dispersion for high-angle annular dark-field scanning transmission electron microscopy (HAADF-STEM) test. Figure R1 gives more images of Re/Ni sample and corresponding elements mapping results. It is seen that the distribution of Re on the Ni substrate is almost monatomic, therefore, it is deduced that Re atoms are atomically dispersed on the Ni balls. Meanwhile, no detected Re-Re bond on Re/Ni sample from Re L₃-edge EXAFS results also support the Re-Ni single atom alloy structure.

In addition, through machine learning method, we have predicted that Re/Ni monoatomic alloy presents excellent catalytic activity and Re is designed as the active species. Moreover, we also performed experiments that Ni balls exhibited almost no catalytic activity, while the methane cracking activity was significantly improved on Re/Ni, which indicated that Re-Ni alloy structure (proved by above comment) is the active site for methane cracking. Hopefully, our following work could validate the active sites behavior during the whole methane cracking process *in situ*, such as by *in-situ* X-ray absorption spectra (XAS), but we regret that we lack such realistic conditions at this moment.

Figure R1. More HAADF-STEM images and elemental mapping results for Re/Ni sample.

As stated previously, this paper is about catalysis – it has nothing to do with industrial hydrogen production, sustainability, or elimination of atmospheric carbon dioxide. It is misleading to readers of Nature to imply this connection. This is an interesting catalysis paper only. Individuals who work in pyrolysis for hydrogen production all understand that the barrier is a low cost reactor system that can efficiently provide and recover heat and remove the carbon at low cost – it is not a catalyst problem it is a heat and mass transfer problem. Nothing of use for individuals working in the field of hydrogen production or sustainable fuels production is learned from this work. It is an interesting and thorough fundamental catalysis paper. All reference of hydrogen production and sustainability should be removed. An appropriate title might be:

Machine Learning Aided Design of Single-atom Alloy Catalysts for Methane Cracking.

Response: Thanks for your feedback. We partially agree with your comments. We have changed the title as suggested and thoroughly revised the manuscript, and

removed the comments about CO₂-free hydrogen production and sustainability, focusing more on catalyst design. Since methane cracking reaction produces hydrogen and carbon as products, the production of hydrogen as the research background is kept only. Some of the changes are as follows:

“The process of CH₄ cracking into H₂ and carbon has gained wide attention for hydrogen production. However, traditional catalysis methods suffer rapid deactivation due to severe carbon deposition. In this study, we discover that effective CH₄ cracking can be achieved at 450 °C over a Re/Ni single-atom alloy via ball milling.”

“Natural gas is abundant and could be converted into hydrogen, playing a significant role in human society. Among various routes of methane conversion, the direct cracking of methane has garnered more and more attention due to its advantages, including high energy conversion efficiency, easy separation of products, and environmental friendliness.”

Given that the other referees seem satisfied with the paper as suitable for Nature, if the above changes are made it might be suitable for publication. I would also suggest that the last sentence of the abstract be reworded to avoid the awkward use of “literatures”.

Response: Thanks for your valuable feedback and constructive suggestions. We have went through the whole manuscript about grammar and spelling and made the necessary changes to the manuscript. The last sentence of the abstract was revised to avoid the awkward use of "literatures." We appreciate your guidance and believe that the revised manuscript meets the publication standards.

“Here, we show the mechanical energy boosts CH₄ conversion clearly and sustained CH₄ cracking over 240 h is achieved, significantly surpassing other approaches in literature.”

Reviewer #2 (Remarks to the Author):

The authors of the manuscript “Accelerated Design of Single-atom Alloy Catalysts for CO₂-free Hydrogen Production from Methane Pyrolysis Aided by Machine Learning” have addressed well enough my suggestion regarding rotating reactor, and explained a bit more in detail the choice of recursive-feature-elimination method for features selection.

With that I do not have any further questions/suggestions.

Response: Thank you for your feedback. We are glad that our revisions were satisfactory. We appreciate your time and input.

Reviewer #2 (Remarks on code availability):

This code requires preinstallation of pymatgen libraries, and it is used for getting the data from a database of surfaces and bulk materials for ab initio calculations. The code itself is written not in a self-explainable manner. However, this is not crucial for the overall work, and in general this code does not contain any unique ideas.

Response: Thank you for your comment. We appreciate your understanding that the code is not self-explanatory but acknowledge that it does not impact the overall work. We will further investigate and improve the code in our future work.